# KIF1C activates and extends dynein movement through the FHF cargo adapter

Ferdos Abid Ali[1,2,5], Alexander J. Zwetsloot [3,5], Caroline E. Stone [1,4], Tomos E. Morgan[1], Richard F. Wademan[1], Andrew P. Carter [1]✉ & Anne Straube [3]✉

Cellular cargos move bidirectionally on microtubules by recruiting opposite polarity motors dynein and kinesin. These motors show codependence, where one requires the activity of the other, although the mechanism is unknown. Here we show that kinesin-3 KIF1C acts as both an activator and a processivity factor for dynein, using in vitro reconstitutions of human proteins. Activation requires only a fragment of the KIF1C nonmotor stalk binding the cargo adapter HOOK3. The interaction site is separate from the constitutive factors FTS and FHIP, which link HOOK3 to small G-proteins on cargos. We provide a structural model for the autoinhibited FTS–HOOK3–FHIP1B (an FHF complex) and explain how KIF1C relieves it. Collectively, we explain codependency by revealing how mutual activation of dynein and kinesin occurs through their shared adapter. Many adapters bind both dynein and kinesins, suggesting this mechanism could be generalized to other bidirectional complexes.

The organization of a cell's content is central to its function. Long-range transport of membrane organelles depends on the opposing microtubule-based motors dynein (cytoplasmic dynein-1) and kinesin (including the kinesin-1 and kinesin-3 families). Almost all cellular cargos move back and forth due to the simultaneous presence of both types of motor, a concept known as bidirectional transport[1–3]. Such an arrangement requires coordination between dynein and kinesin to prevent unproductive stalling and tug of war. A large body of evidence shows that opposite polarity motors also depend on each other during motility; diminishing either dynein or kinesin-1 activity leads to predominantly static organelles, lipid droplets and messenger RNA–protein complexes[4–11]. This phenomenon, referred to as codependence of antagonistic motors[2], also occurs between dynein and kinesin-3s. For example, depletion of KIF1C reduces transport of integrin-containing vesicles to the same extent in both directions and the retrograde transport of dense core vesicles is perturbed when the *Caenorhabditis elegans* KIF1A homolog Unc-104 is mutated[12,13]. Although it is well established that dynein and kinesin require each other on cargos, the molecular mechanism of codependence remains unclear.

## Results

### Dynein and KIF1C form codependent complex bridged by HOOK3

Human cytoplasmic dynein is a weakly processive motor on its own[14], requiring the cofactor dynactin and a coiled-coil cargo adapter for full activation[15–17]. Many of dynein's activating cargo adapters can also bind kinesins, including BICD2 (refs. 18,19), BICDR1 (refs. 20,21), HOOK[21–25], TRAK[26–28] and JIP3 (refs. 29–31). HOOK3 is of particular interest as it has been shown to bind dynein/dynactin and the kinesin-3 motor KIF1C simultaneously in vitro[22]. To determine directly whether these opposite polarity motors show codependence, we reconstituted purified dynein–dynactin–HOOK3 (DDH) complexes bound to KIF1C and analyzed their motile behavior using single-molecule total internal reflection fluorescence (TIRF) microscopy assays (Fig. 1a).

In the presence of all components, complexes undertook processive minus end- (dynein driven, 84.4%) or plus end-directed (kinesin driven, 12.8%) motility (Fig. 1b,c). The observation of predominant minus end runs agrees with a previous study on this cocomplex[22]. In addition, we observe occasional switches in direction during runs

[1]MRC Laboratory of Molecular Biology, Cambridge, UK. [2]School of Biochemistry, University of Bristol, Bristol, UK. [3]Centre for Mechanochemical Cell Biology and Warwick Biomedical Sciences, Warwick Medical School, University of Warwick, Coventry, UK. [4]John Innes Centre, Norwich Research Park, Norwich, UK. [5]These authors contributed equally: Ferdos Abid Ali, Alexander J. Zwetsloot. ✉e-mail: cartera@mrc-lmb.cam.ac.uk; anne@mechanochemistry.org

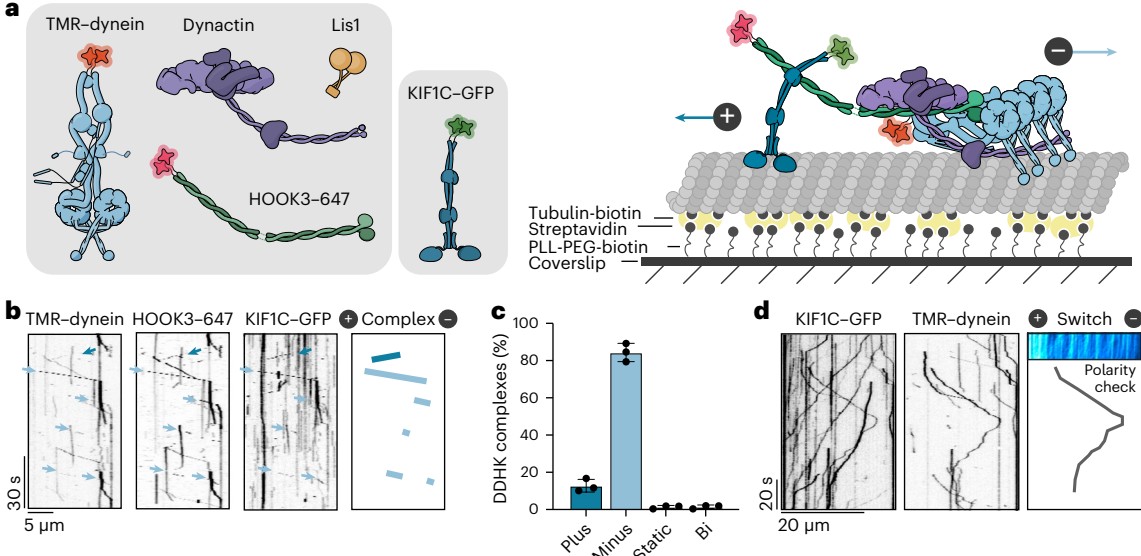

**Fig. 1 | Dynein and KIF1C form coordinated cocomplexes scaffolded by HOOK3. a**, Left: the schematics of proteins used in reconstitution of cocomplexes. Dynein and HOOK3 are labeled via N-terminal SNAP tags with TMR and Alexa-647, respectively. KIF1C is fused to a C-terminal eGFP. Right: representation of single-molecule motility assay used in this study. The microtubules are immobilized by sandwiching streptavidin between tubulin-biotin and PLL-PEG-biotin on the glass coverslip. The labeled motor proteins and adapters are then added and their motility imaged. **b**, Kymographs from TIRF images showing colocalized movement of dynein, HOOK3 and KIF1C.

Right: minus end runs of DDHK complexes in light blue and plus end runs in dark blue. **c**, Quantification of DDHK complex directionality, with those undertaking directional reversals indicated as bidirectional (Bi). A total of $n = 1{,}418$ motor complexes were used across three separate experiments ($n = 190$ plus directed, $n = 1{,}182$ minus directed, $n = 22$ static and $n = 24$ bidirectional). The data are presented as mean values ± s.d. **d**, An example of a directional switch of a DDHK complex during a run. Inset: polarity check with KIF1C–GFP only, which was performed after every experiment containing both KIF1C and dynein to establish microtubule polarity.

(1.5%) (Fig. 1c,d), reminiscent of bidirectional cargo movement in the cell[24]. This suggests that coupling opposite polarity motors via their adapters supports processive unidirectional transport and prevents an unproductive tug of war. This differs from artificial, DNA-linked systems, where motors are in a force balance and resist each other's movement so that the cocomplexes are primarily static[32–34].

We noticed a significant increase in the dynein landing rate in the presence of full-length KIF1C ($K_{FL}$) (Fig. 2a,b), an effect that has not been previously described. This is not because of direct KIF1C–dynein binding but is rather mediated through HOOK3 as omission of this adapter abolished dynein activity (Extended Data Fig. 1). To dissect the mechanism of this activation, we focused on a KIF1C stalk ($K_S$) construct that does not contain the motor domain but includes the region required to bind HOOK3 (ref. 23) (GST–KIF1C$^{642–922}$–green fluorescent protein (GFP)). Strikingly, recruitment of DDH to microtubules increases 2.6-fold in the presence of the KIF1C stalk construct (Fig. 2a,b), indicating that this region is sufficient for binding HOOK3 and activating the cocomplex. A similar activating effect was observed vice versa: KIF1C landing rate on microtubules increased twofold in the presence of dynein and dynactin (dynein, dynactin, HOOK3 and $K_{FL}$–GFP (DDHK$_{FL}$)) compared with HOOK3–KIF1C (HK$_{FL}$), even if the motor domain was omitted by using a dynein tail construct ($D_T$) (Fig. 2c,d). Dynactin alone is also sufficient to increase the landing rate, although to a lesser extent (1.4-fold compared with HK$_{FL}$) (Fig. 2c,d). Collectively, our results show that KIF1C and dynein can facilitate each other's microtubule binding activity using their nonmotor domains.

## FHF is an autoinhibited cargo adapter

To understand how this mutual activation is mediated, we investigated the regulation of the cargo adapter. Unlike other cargo adapters, HOOKs form a stable assembly with two other accessory factors, FTS and FHIP (Extended Data Fig. 2a)[35,36]. The proposed binding sites of KIF1C, FTS and FHIP1B to HOOK3 are all located within the adapter's C-terminal 165 residues[22,35]. To determine the precise molecular organization

of the FTS–HOOK–FHIP (FHF) complex, we used cryogenic electron microscopy (cryo-EM). Our 3.2 Å reconstruction of full-length FTS and FHIP1B in complex with a C-terminal HOOK3 fragment (residues 571–718) allowed an atomic model to be built for each subunit (Fig. 3a,b, Table 1 and Extended Data Figs. 2b and 3). In the absence of HOOK3 the complex between FTS and FHIP1B does not form (Extended Data Fig. 2c), suggesting that HOOK3 is central to FHF assembly (Fig. 3b). Our structure reveals HOOK3's interactions involve the last of its four coiled-coil domains (CC4) and a C-terminal extension (CTE), consisting of a flexible region followed by a single alpha helix (residues 694–702) (Fig. 3b and Extended Data Fig. 3g). The first contact point (site 1) involves one of HOOK3's two CTEs, with the helix latching onto a hydrophobic pocket within FHIP1B. Site 2 comprises a span of HOOK3 CC4 residues (653–688), packing tightly against the catalytically dead E2 ubiquitin ligase domain in FTS (Fig. 3b and Extended Data Fig. 3h). At site 3, the opposite side of HOOK3 CC4 (spanning residues 663–670) binds a C-terminal alpha helix on FTS. This helix is situated within a long loop (FTS 'belt') that wraps around HOOK3 before docking into at the midzone of FHIP1B (Fig. 3b and Extended Data Fig. 3h,i). Together, these interactions mean that the extreme C-terminus of HOOK3 is buried within FHIP1B and FTS and cannot interact with cargo directly. Instead, the opposing face of FHIP1B comprises a highly conserved patch of surface-exposed residues (Extended Data Fig. 4a). AlphaFold2 (AF) suggests this is a binding site for small G-protein Rab5, which has been shown to link FHF to early endosomes (Fig. 3c and Extended Data Fig. 4b,c)[36,37]. The location of this site means the adapter is positioned to be extended away from the cargo.

All HOOK family members (HOOK1–3) bind FTS due to their highly conserved EEKxxxSAWYN motif (Fig. 3d). This motif is present on both sides of the coiled coil and interacts with FTS at sites 2 and 3. Consistent with its importance, mutations in it abolish HOOK1 binding to FTS[35].

In contrast to FTS, there is evidence that HOOK homologs show preferences for different FHIPs (FHIP1A, FHIP1B, FHIP2A and FHIP2B)

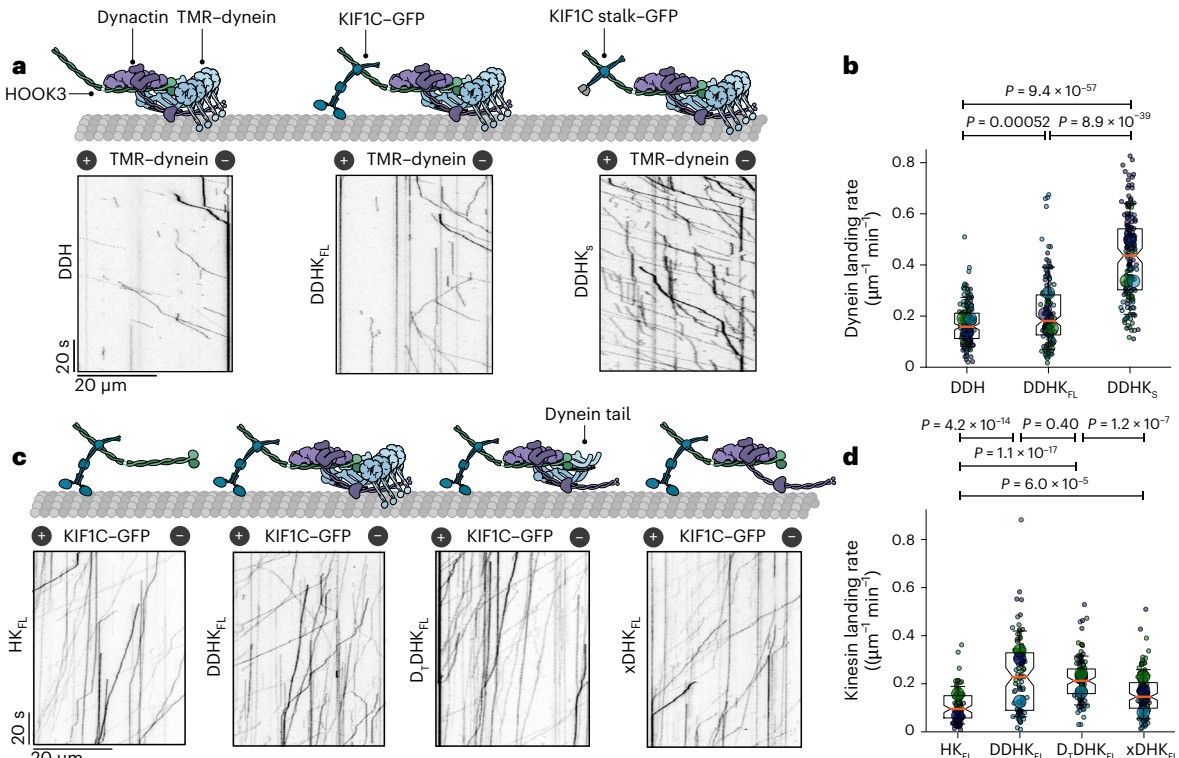

**Fig. 2 | Dynein and KIF1C are codependent in vitro. a**, Schematics (top) and representative kymographs (bottom) from TIRF imaging of TMR-labeled DDH; $DDHK_{FL}$; or dynein, dynactin, HOOK3 and GST–KIF1C stalk–GFP ($DDHK_S$). All experiments were performed in the presence of Lis1. **b**, A superplot of the TMR–dynein landing rate on microtubules for different cocomplex combinations (as in **a**). The small dots indicate single microtubules and the large dots indicate experimental averages. $n = 165$ for DDH, $n = 150$ for DDHK and $n = 171$ for $DDHK_S$, where $n$ is number of microtubules over which the landing rate was measured. The boxes show quartiles with whiskers spanning 10–90% of the data, and the median is highlighted by the orange line. The exact $P$ values shown above the graphs were calculated using the Kruskal–Wallis $H$ test followed by a Conover's post hoc test to evaluate pairwise interactions with a multiple comparison correction applied using Holm–Bonferroni. **c**, Schematics (top)

and representative kymographs (bottom) from TIRF imaging of KIF1C–GFP mixed with HOOK3 ($HK_{FL}$), dynactin, HOOK3 and full-length dynein ($DDHK_{FL}$), dynein tail, dynactin, HOOK3 and KIF1C–GFP ($D_TDHK_{FL}$) or dynactin, HOOK3 and KIF1C–GFP ($xDHK_{FL}$). **d**, A superplot of the KIF1C–GFP landing rate on microtubules for different cocomplex combinations (as in **c**). The small dots indicate single microtubules, and the large dots indicate experimental averages. $n = 94$ for $HK_{FL}$, $n = 97$ for $DDHK_{FL}$, $n = 113$ for $D_TDHK_{FL}$ and $n = 138$ for $xDHK_{FL}$, where $n$ is number of microtubules over which landing rate was measured. The boxes show the quartiles with whiskers spanning 10–90% of the data, and the median is highlighted by the orange line. The exact $P$ values shown using the graphs using the Kruskal–Wallis $H$ test followed by a Conover's post hoc test to evaluate pairwise interactions with a multiple comparison correction applied using Holm–Bonferroni.

in cells[36]. Surprisingly, the CTE helix, found at site 1 and forming the main contact between HOOK3 and FHIP1B, is also highly conserved between HOOK family members (Fig. 3d). This suggests that HOOK–FHIP specificity is mediated either by the preceding CTE loop, which varies in sequence and length among HOOKs, or by another mechanism such as specific cotranslation of selected messenger RNAs[38]. To test the former model, we coexpressed HOOK2 or HOOK3 with FTS and different FHIPs (FHIP2A or FHIP1B) (Fig. 3e). HOOK2 was previously reported to prefer FHIP2A and HOOK3 to preferentially bind FHIP1B[36]. Unexpectedly, we found that all combinations of HOOKs and FHIPs produced stable FHF complexes (Fig. 3f,g). This indicates that the FHF specificity previously observed in proximity-dependent proteomics experiments is probably driven by a cell-based mechanism and that the structural basis for FHF complex formation is universal to all FHIP homologs.

Factors that bind the C-terminus of other adapters, such as BICD2 and Spindly, are thought to activate them[39–42]. To investigate whether FTS–FHIP1B binding to the C-terminus activates HOOK3, we tested the ability of FHF to facilitate processive dynein movement in single-molecule motility assays. Unexpectedly, we found that FHF was 11-fold less efficient in activating dynein motility than a truncated HOOK3 construct containing only the N-terminal 522 amino acids ($DDH^{1–522}$) ($0.065 \pm 0.071$ versus $0.75 \pm 0.44$ average processive events per micrometer per second, respectively) (Fig. 4a,b). FHF did

not significantly increase the frequency of motile events relative to dynein–dynactin alone ($0.020 \pm 0.044$ processive events per micrometer per second) (Fig. 4b). However, the small number of DDFHF complexes that engage with microtubules have run lengths that are similar to those formed with the constitutively active, truncated HOOK3 ($6,347 \pm 5,895$ versus $5,242 \pm 1,808$ nm, respectively) (Fig. 4c). These data suggest that whereas FHF can occasionally activate dynein, it is largely unable to do so and therefore FTS–FHIP1B binding does not activate HOOK3. Strikingly, adding KIF1C stalk led to a ~3.6-fold increase in processive events ($0.26 \pm 0.20$ processive events per micrometer per second) compared with DDFHF (Fig. 4a,b), indicating that KIF1C stalk activates HOOK3 also in the context of FHF.

To understand the FHF autoinhibition at the structural level, we used crosslinking mass spectrometry and mapped the interactions within the full-length molecule. We found numerous crosslinks between the HOOK3 coiled-coil domains, with CC2 crosslinking to CC3 and CC4 (Extended Data Fig. 5a). We mapped these crosslinks onto a model of full-length FHF, constructed by combining our cryo-EM model with three different and partially overlapping AlphaFold2-predicted segments of HOOK3 (Extended Data Fig. 5b). Collectively, this shows an N- to C-terminal interaction with an 'elbow' between CC2 and CC3 that enables a fold-back conformation to be adopted, which would be sterically incompatible with dynein–dynactin binding (Fig. 4d and Extended Data Fig. 5c)

## KIF1C stalk opens FHF by binding a regulatory site on HOOK3

Next, we addressed how the KIF1C stalk binds HOOK3 and interplays with FTS and FHIP1B at the molecular level. KIF1C is a stable dimer consisting of an N-terminal motor domain and a C-terminal tail. To obtain a full-length predicted structure of KIF1C, we stitched seven AlphaFold2 models together (Extended Data Fig. 6a). This prediction suggests there are three coiled-coils (called CC1–3 here) interspersed with two globular domains. The kinesin-3-specific forkhead-associated domain sits between CC1 and CC2. The second globular region, between CC2 and CC3, was not previously known to form a structured domain. Here, we refer to it as the stalk globular domain (SGD) (Fig. 5a,b). The KIFC stalk we used in our experiments encompasses CC2, the SGD and CC3.

This KIF1C stalk forms a stable complex with FHF over gel filtration, indicating the binding sites for FTS–FHIP1B and KIF1C stalk do not overlap on HOOK3 (Extended Data Fig. 7a). We used mass photometry to measure the molecular weight of the complex as ~434 kDa, which is close to the predicted molecular weight of one FHF and one dimeric KIF1C stalk (430 kDa) (Extended Data Fig. 7b).

To map the binding sites more precisely, we used cryo-EM on a KIF1C stalk construct bound to FTS–FHIP1B and a C-terminal HOOK3 encompassing residues 451–718 (FH^CF) (Extended Data Fig. 7c and Table 1). We observed extra density corresponding to the KIF1C stalk at the hinge between HOOK3 CC3 and CC4 (Fig. 5c), although the resolution was low due to flexibility of this part of HOOK3 (Extended Data Fig. 6b and 8a–c). We therefore used AlphaFold2 to screen pairwise interactions of sequential dimeric KIF1C fragments (in 120-residue increments) against a dimer of HOOK3 encompassing CC3 and CC4 (Extended Data Fig. 8d). This identified part of KIF1C's SGD and CC3 as the top hit (residues 722–840). The resulting AlphaFold2 model showed KIF1C's SGD binding HOOK3 at a position that matches the extra density in our cryo-EM structure (Fig. 5c,d). In further support of this assignment, deleting residues 722–840 from the KIF1C stalk abolished its interaction with HOOK3 (Extended Data Fig. 7d). The inverse is also true as FHF with a truncated HOOK3 (comprising residues 629–718), which lacks the kinesin binding site, no longer bound to KIF1C stalk (Extended Data Fig. 7e). Furthermore, crosslinking mass spectrometry revealed numerous crosslinks from the SGD domain to the HOOK3 CC3–CC4 interface (Fig. 5e and Extended Data Fig. 8e). Overall, our mapping studies showed that the KIF1C stalk mainly binds FHF within HOOK3 (residues 607–632), which is nonoverlapping with the FTS and FHIP1B binding sites that cover HOOK3 residues 663–718.

We observed that KIF1C binding to HOOK3 overlaps with the intramolecular interaction region between HOOK3's CC2 and CC3–CC4

(Fig. 4d). Therefore, KIF1C binding might disrupt the autoinhibited form of FHF (Fig. 5f). To verify this directly, we split HOOK3 at its elbow and expressed H^N (HOOK3 amino acids 1–450) and strep-tagged FH^CF (containing strep-HOOK3 451–718) separately. In pulldown experiments, FH^CF bound H^N, suggesting they can interact, in line with our crosslinking mass spectrometry data (Fig. 5g,h, lane 3). When we added KIF1C stalk (K_S), we found it reduced H^N binding to FH^CF to a similar level as background binding of the H^N only control (Fig. 5g,h, lanes 1 and 4). Furthermore, unlike the KIF1C stalk, which readily binds FH^CF (Fig. 5g,h, lane 6), a KIF1C stalk lacking the HOOK3 binding site (K_S^ΔSGD) was not able to bind FHF or displace H^N (Fig. 5g,h, lane 5). Together, these data suggest that KIF1C stalk can shift the equilibrium toward an open configuration of HOOK3 and relieve FHF autoinhibition. This

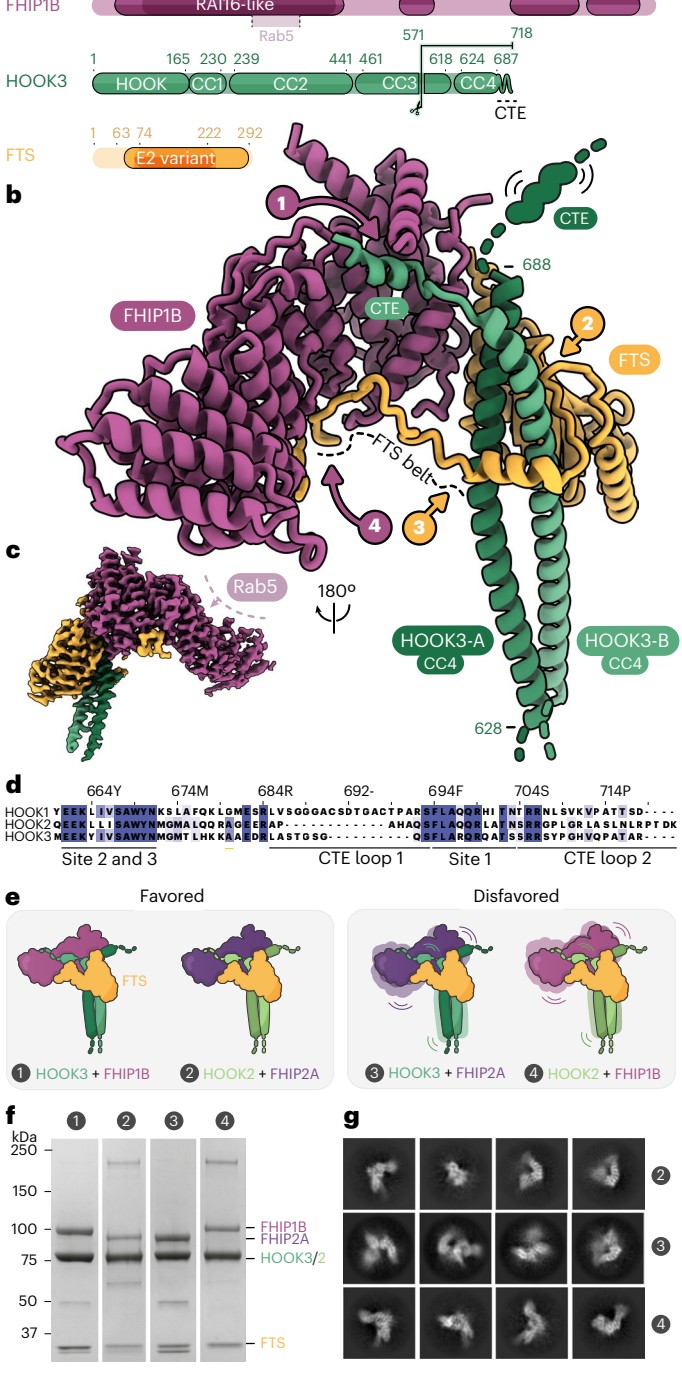

**Fig. 3 | HOOK anchors FHIP and FTS in FHF complexes. a**, The domain architecture of FHIP1B, HOOK3 and FTS. A putative Rab5 binding site is indicated in FHIP1B. The scissor cut at residue 571 in HOOK3 indicates the truncation site used for FHF cryo-EM construct. **b**, An atomic model of FHF complex built from 3.2 Å cryo-EM structure, highlighting the four main interaction sites and the flexible regions within HOOK3. **c**, Cryo-EM density of FHF with the AlphaFold2-predicted Rab5 binding site highlighted on FHIP1B (see Extended Data Fig. 4 for details). **d**, Sequence conservation between human HOOK1, HOOK2 and HOOK3. The residue numbers refer to HOOK3. **e**, The combinations of FHF coexpressed and purified to assess complex formation. FTS–HOOK3–FHIP1B (complex 1) and FTS–HOOK2–FHIP2A (complex 2) were classed as favored assemblies and FTS–HOOK3–FHIP2A (complex 3) and FTS–HOOK2–FHIP1B (complex 4) as disfavored assemblies based on ref. 36. **f**, SDS–PAGE of purified FHF (complexes 1–4) show that HOOK2 or HOOK3 promiscuously bind FHIP2A or FHIP1B in vitro. This purification was performed once as shown in **f** with the full-length HOOK proteins bound to FTS–FHIP1B or FTS–FHIP2A and once with C-terminal fragments of HOOK2 (residues 566–719) and HOOK3 (571–718) bound to FTS–FHIP1B/FHIP2A, for which cryo-EM data were collected in **g**, revealing the same HOOK–FHIP pattern of binding both times. **g**, Cryo-EM 2D class averages of favored (2) and disfavored (3 and 4) complexes (using HOOK2^566–719 and HOOK3^571–718 fragments coexpressed with FTS–FHIP1B or FTS–FHIP2A), showing recognizable views in all cases and indicating stable FTS–HOOK–FHIP assembly in all combinations tested.

**Table 1 | Cryo-EM and refinement statistics**

| | FH$^{571–718}$F | FH$^{571–718}$F | FH$^{451–718}$F+KIF1CS$^{674–822}$ | FH$^{451–718}$F+KIF1CS$^{674–922}$ | FH$^{451–718}$F+KIF1CS$^{674–922}$ |
|---|---|---|---|---|---|
| Electron Microscopy Data Bank ID | 18302 | | 18303 | — | — |
| Protein Data Bank ID | 8QAT | | — | — | — |
| Data collection and processing | | | | | |
| Magnification | 81,000× | 75,000× | 81,000× | 81,000× | 81,000× |
| Voltage | 300 | 300 | 300 | 300 | 300 |
| Pixel size (Å) | 1.059 | 1.09 | 1.059 | 1.059 | 1.059 |
| Electron exposure (e$^-$ Å$^{-2}$) | 44 | 34 | 44 | 43 | 54 |
| Defocus range (µm) | 1.6–2.6 | 1.6–2.6 | 1.6–2.6 | 1.3–2.7 | 1.3–2.7 |
| Initial movies | 26,383 | 16,446 | 11,839 | 5,726 | 7,424 |
| Initial particle images | 902,151 | 1,605,779 | 2,368,723 | 1,297,609 | 952,300 |
| Final particle images | 273,204 | | | 507,489 | — |
| Symmetry imposed | C1 | C1 | C1 | C1 | C1 |
| Map resolution (Å) | 3.2[a] | | | 4.8[b] | |
| Fourier shell correlation threshold | 0.143[a] | | | 0.143[b] | |
| Map-sharpening B factor (Å$^2$) | −116.705[a] | | | −10[b] | |
| Model refinement | | | | | |
| Model resolution (Å) | 3.52[a] | | | — | — |
| Fourier shell correlation threshold | 0.5[a] | | | — | — |
| Model composition | | | | | |
| Nonhydrogen atoms | 7,758[a] | | | — | — |
| Protein residues | 970[a] | | | — | — |
| Ligands | — | | | — | — |
| Mean B factors (Å$^2$) | | | | | |
| Protein | 100.08[a] | | | — | — |
| Ligand | — | | | — | — |
| Root mean square deviations | | | | | |
| Bond lengths (Å) | 0.006[a] | | | — | — |
| Bond angles | 1.267°[a] | | | — | — |
| Validation | | | | | |
| Clash score | 3.47[a] | | | — | — |
| MolProbity score | 1.14[a] | | | — | — |
| Rotamers outliers (%) | 0.35[a] | | | — | — |
| Ramachandran plot | | | | | |
| Favored (%) | 98.01[a] | | | — | — |
| Allowed (%) | 1.89[a] | | | — | — |
| Outliers (%) | 0.1[a] | | | — | — |

[a]Value obtained from three merged datasets. [b]Value obtained from two merged datasets.

would facilitate dynein–dynactin recruitment and formation of the KIF1C–dynein cocomplex.

### KIF1C acts as a processivity factor for dynein
Finally, we asked why cocomplex formation might be advantageous for KIF1C and dynein. We hypothesized that the motors could be passive passengers to efficiently relocate to the opposite microtubule end or, alternatively, could directly affect each other's movement. To test this, we compared the motility parameters of dynein and kinesin with and without the opposing motor present (Fig. 6).

In the case of KIF1C–HOOK3, the presence of the dynein machinery led to a moderate decrease in both the speed and run length (Fig. 6a–c). However, the duration motor complexes stay on the microtubule ('dwell time') was unchanged (Fig. 6d). The effect we observed required the presence of the motor domains, which suggests that dynein is engaged with the microtubule during KIF1C-driven runs.

Analysis of DDH motility revealed that the presence of $K_{FL}$ decreased the velocity of minus end runs (Fig. 6e,f) but increased both the run length and microtubule dwell time (Fig. 6g,h). These effects were not seen for the KIF1C stalk construct (Fig. 6f,h) showing that they required the KIF1C motor domain. These data suggest that, in addition to engaging with the microtubule, KIF1C is able to act as a processivity factor for dynein. However, despite interacting with

the microtubules, neither KIF1C nor dynein cause stalling during opposite polarity runs when they are coupled via HOOK3 (Extended Data Fig. 9).

## Discussion

### A general mechanism for codependence?

Codependence of microtubule motors, where the inhibition or removal of one motor results in reduced cargo transport in both directions, was observed in many cellular studies[1–13]. Three models were proposed to explain this phenomenon: mechanical activation (opposing forces open up and activate motors), steric disinhibition (interaction of motors with each other releases their autoinhibition) and microtubule tethering (binding by both motors allows cargos to stay attached to their tracks)[2]. Our data provide evidence that dynein and kinesin can relieve each other's inhibition to increase the frequency of processive movements on microtubules (steric disinhibition model). Originally, this model envisioned that both motors directly interact[2]. Here, we propose instead that disinhibition occurs through regulating the scaffolding adapter HOOK3.

Our observations that motorless kinesin and dynein constructs activate the opposing motor suggests that mechanical activation is not required for the initiation of transport in this system but that it depends on the interactions with the HOOK3 cargo adapter. When the motor domains of the opposing dynein or KIF1C are present, we observe moderate reductions in speed providing evidence that both motors are simultaneously engaged with the microtubule. This additional microtubule interaction could act as a tether and reduce the unbinding rate of the complex. Consistent with this idea, KIF1C increases dynein processivity. However, this effect is not reciprocal as the duration of KIF1C runs is independent of dynein motors.

Although not yet formally tested, our described steric disinhibition mechanism of motor codependence may be applicable to other adapter-scaffolded ensembles. For example, KIF1C has also been shown to bind to the dynein–dynactin adapters BICDR1 (ref. 20) and BICD2 (ref. 43). The interaction is thought be with the adapter C termini akin to the situation with HOOK3 (refs. 20,43). In addition to KIF1C, several other kinesins interact with dynein–dynactin adapters. BioID suggests the kinesin-3 motor, KIF1B, also binds FHF complexes[36], although its exact interaction site is not known. There is evidence that kinesin-1 motors interact with the dynein adapters TRAK2 (refs. 26–28), JIP3 (refs. 29,31,44), BICD2 (refs. 19,45) and HAP1 (ref. 46). It was recently shown in a lysate-based system that perturbation of kinesin-1 or dynein–dynactin components reduced TRAK2 dependent motility of the opposite motor[28]. This is consistent with kinesin-1 activating dynein via its scaffolding adapter (and vice versa). However, the details of the activation may differ from KIF1C–HOOK3 as the mapped binding sites for kinesin on TRAK and other adapters such as JIP3, overlap with those of dynein–dynactin.

### Kinesin and dynein stabilize the active conformation of FHF

Many adapters use their C termini to directly bind small G-proteins, which tether them to their membrane cargos and activate them[47–49]. HOOK3, in contrast, is likely to bind its small G-protein (Rab5) via a site on FHIP1B that is well separated from the coiled coil. This suggests that FHF might be recruited to its endosome cargo[24,25,37,50] in its inhibited conformation (Fig. 7a).

We propose that KIF1C's stalk can lead to a partial opening of HOOK3 that allows dynein–dynactin recruitment (Fig. 7b). We previously described that KIF1C is itself autoinhibited by interactions between its motor domain and the stalk (including the HOOK3 binding region in KIF1C)[23] (Fig. 7a). This explains why the KIF1C stalk alone is more efficient in activating HOOK3 than $K_{FL}$. However, as the KIF1C stalk does not induce DDFHF motility to the same level as the constitutively active HOOK3 construct (HOOK3[1–522]), it is possible that other,

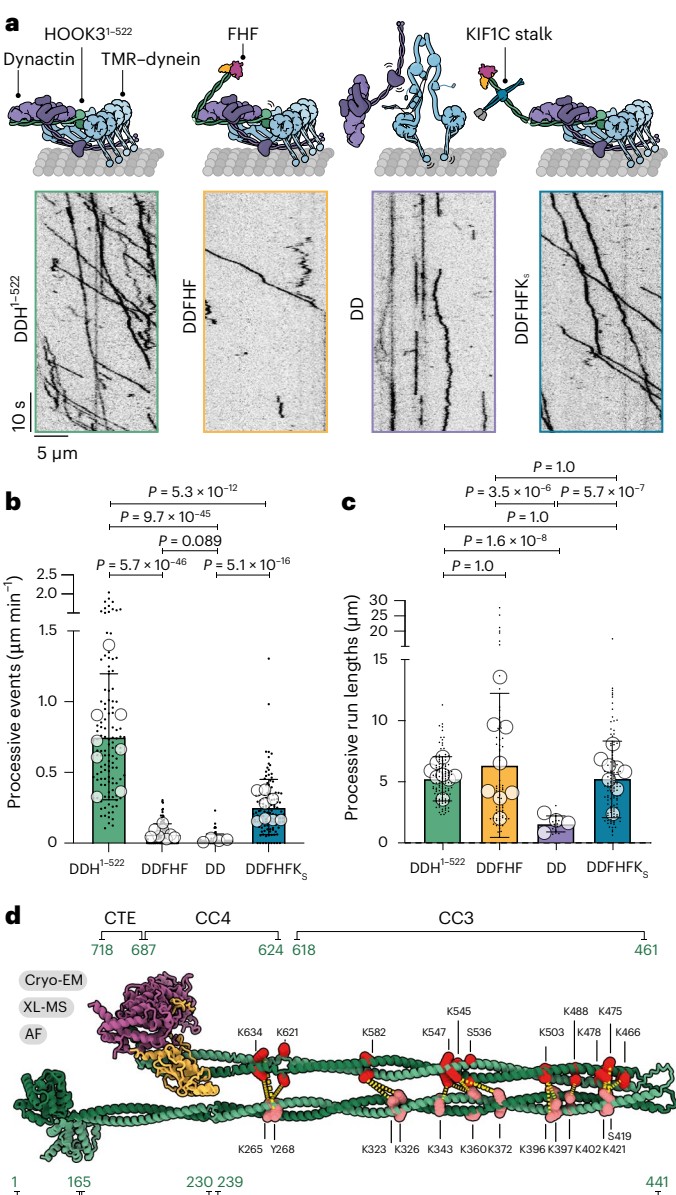

**Fig. 4 | FHF is an autoinhibited cargo adapter. a**, Representative kymographs of TMR–dynein and dynactin in presence of either HOOK3[1–522] (left), full-length FHF (middle left), no other factor (DD, middle right) or full-length FHF + KIF1C stalk (right). All the samples also included Lis1. **b,c**, Quantification of processive events (**b**) and run lengths (**c**) from eight technical replicates (apart from DD condition, which had four technical replicates). The bars indicate the mean ± s.d., the small dots indicate data for each microtubule and the large dots indicate experimental averages. Statistics were performed on total microtubules, which is $n = 120$ microtubules for all conditions apart from the DD condition, where $n = 60$. A total of 15 microtubules were counted per replicate for all conditions. The exact $P$ values shown above the graphs were calculated using the Kruskal–Wallis nonparametric test with Dunn's multiple comparison. **d**, Full-length FHF composite model derived from stitched AF predictions and the cryo-EM structure of HOOK3 624–718, FTS and FHIP1B showing DSSO crosslinks (yellow dashes) between HOOK3 N-terminal residues (salmon orange) and HOOK3 C-terminal residues (red) identified using crosslinking mass spectrometry (XL-MS).

as yet unidentified, factors are required for full adapter opening and activation.

In this study we focused on KIF1C's activation of dynein as the majority of the movement we observed for the cocomplex was minus end directed

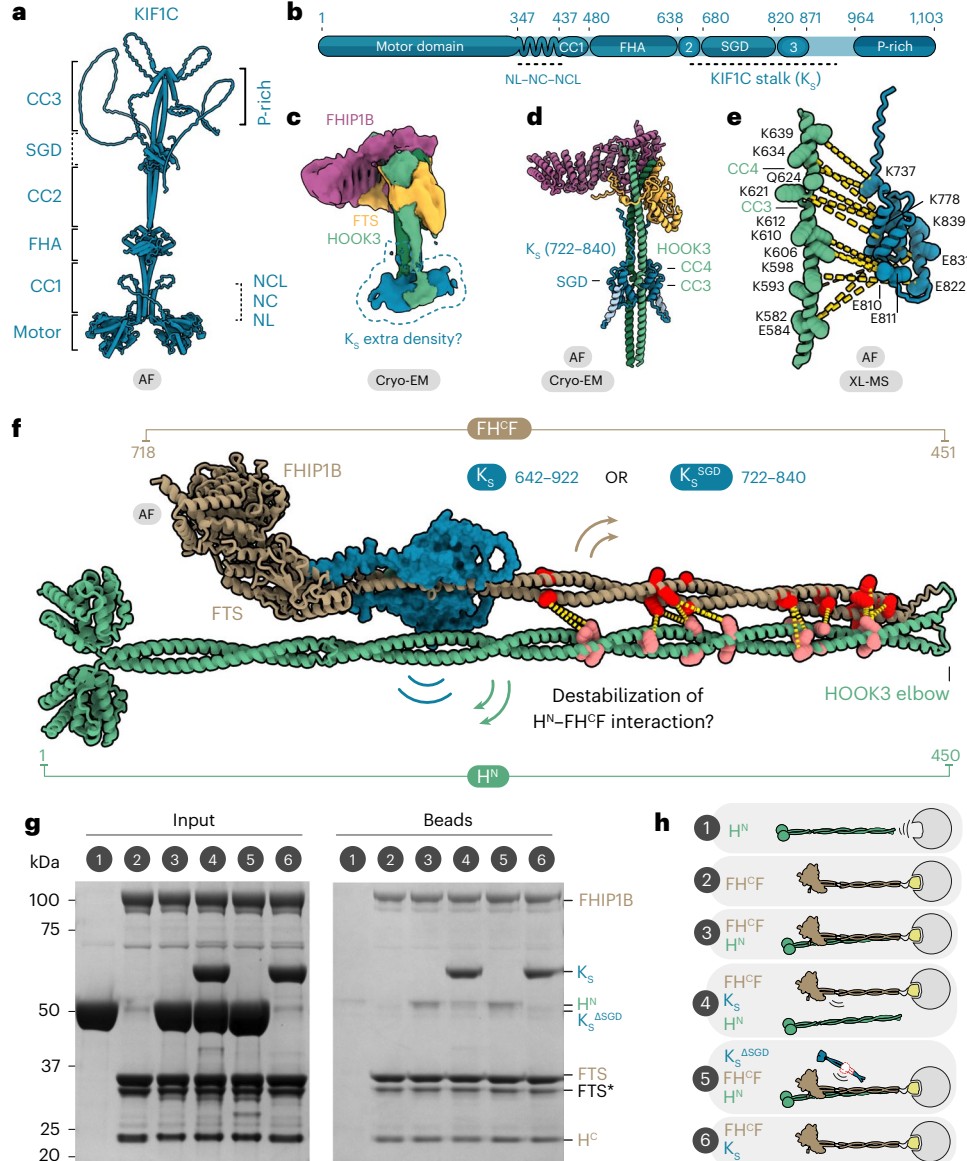

**Fig. 5 | KIF1C stalk binds HOOK3 away from the FTS–FHIP1B site and relieves adapter autoinhibition. a**, A stitched structural model of the AF-predicted $K_{FL}$ molecule. NL, neck linker; NC, neck coil; NCL, neck coil linker; FHA, forkhead-associated domain; P-Rich, proline-rich region. **b**, Domain architecture of KIF1C with the KIF1C stalk region highlighted below the main bar. **c**, The segmented cryo-EM density of FHF bound to a shorter KIF1C stalk (674–922, the dashed blue region), shown at a low-threshold contour level. **d**, AF model of HOOK3 residues 571–718 bound to KIF1C residues 722–840 ($K_S^{722-840}$). Note that FTS, FHIP1B and HOOK3 (627–705) models from the cryo-EM structure are superposed on the HOOK3$^{571-718}$–$K_S^{722-840}$ prediction to give a composite structure. **e**, Expansion of AF model in **d** showing one copy of HOOK3 at the CC3–CC4 junction bound

to one copy of $K_S^{722-840}$. The yellow dashes indicate crosslinks observed by XL-MS between the two proteins using DSSO crosslinker. **f**, A representation of a putative steric clash that would occur upon the SGD of KIF1C stalk latching onto the HOOK3 C terminus. $H^N$, HOOK3 N terminus 1–450; $K_S$, GST–KIF1C stalk (residues 642–922) and $K_S^{SGD}$, GST–SGD (residues 722–840). **g**, Input and bead samples from the pulldown experiment where strep-$FH^CF$ was used as bait. The key for the sample number above the gel is given in **h**. The experiment was repeated twice. **h**, Schematics of the included proteins and the presumed complexes formed. The yellow square shape in $FH^CF$ cartoons represents the strep tag, and the circle shape represents the bead.

and because the network of interactions between KIF1C and FHF were previously elusive. However, we suggest activation of KIF1C movement by dynein may occur in the same way (Fig. 7c). In line with this, structural modeling shows the dynein–dynactin complex would clash with the inhibited form of FHF (Extended Data Fig. 5c), suggesting it can stabilize the open form in which HOOK3's C terminus is exposed for KIF1C binding and activation (Fig. 7c). This explains the higher KIF1C–HOOK3 recruitment to microtubules in the presence of both the full and partial dynein machinery (Fig. 2c,d). We also reported that the pointed end of dynactin alone can bind the adapter[51], which could explain the ability of dynactin alone to moderately activate KIF1C–HOOK3 binding to microtubules.

## What is the benefit of dynein–kinesin cocomplexes?

A key feature of reciprocal motor activation is that it ensures the presence of both dynein and kinesin simultaneously on cargoes before transport occurs. This, in turn, can have a number of advantages. First, it provides an efficient mechanism for motor recycling so that they are available at their respective start points[52–56]. Second, the nondriving motor can act as a processivity factor to allow longer distance transport. Here, we report direct evidence for KIF1C increasing the processivity of dynein–dynactin due to its ability to weakly interact with the microtubule while being transported backward. Third, simultaneous recruitment of dynein and kinesin means the nondriving motor can respond

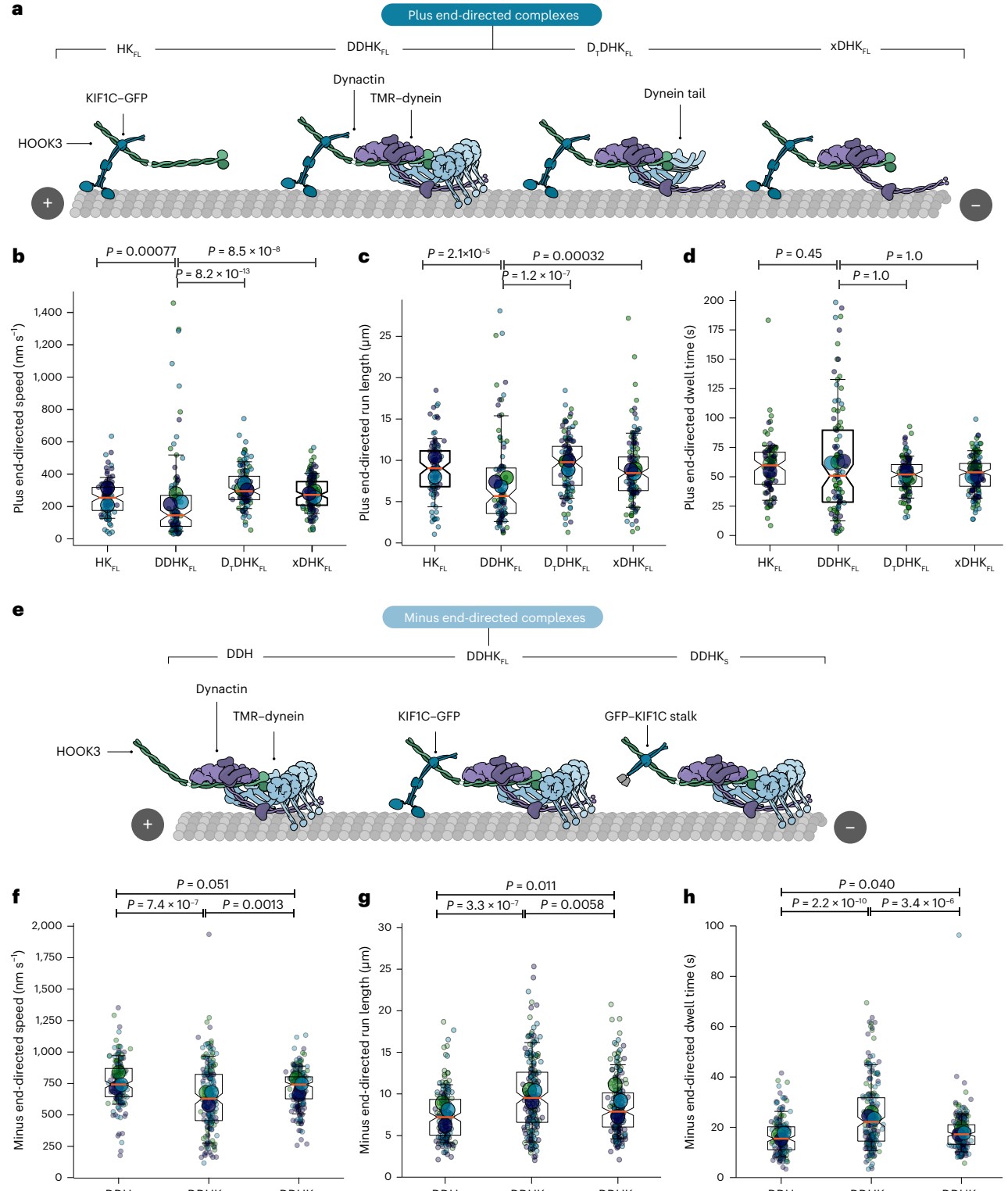

**Fig. 6 | Motility behavior of dynein and kinesin-driven cocomplexes. a**, Cartoons depicting the proteins used in each plus end-directed cocomplex. **b–d**, Superplots showing plus end-directed speed (**b**), run lengths (**c**) and dwell time (**d**) of KIF1C–HOOK3 complexes (HK$_{FL}$) in the presence of dynein and dynactin (DDHK$_{FL}$), dynein tail and dynactin (D$_T$DHK$_{FL}$) and dynactin only (xDHK$_{FL}$). $n$ = 94, 95, 113 and 137, respectively. The small dots show data per microtubule analyzed ($n$) and the large dots show experimental averages. The boxes show quartiles, whiskers show 10–90% of data and the median is highlighted by the orange line. The exact $P$ values shown above graphs using Kruskal–Wallis $H$ test followed by a Conover's post hoc test to evaluate pairwise interactions with a multiple comparison correction applied using Holm–Bonferroni. **e**, Cartoons depicting

the proteins used in each minus end-directed cocomplex. **f–h**, Superplots showing minus end-directed speed (**f**), run lengths (**g**) and dwell time (**h**) of DDH complexes in the presence of either KIF1C full-length (DDHK$_{FL}$) or KIF1C stalk (GST–KIF1C stalk–GFP, DDHK$_S$). $n$ = 165, 197 and 199, respectively. The small dots show data per microtubule analyzed ($n$) and large dots show experimental averages. The boxes show the quartiles, the whiskers show 10–90% of data and the median is highlighted by the orange line. The exact $P$ values shown above the graphs were calculated using the Kruskal–Wallis $H$ test followed by a Conover's post hoc test to evaluate pairwise interactions with a multiple comparison correction applied using Holm–Bonferroni.

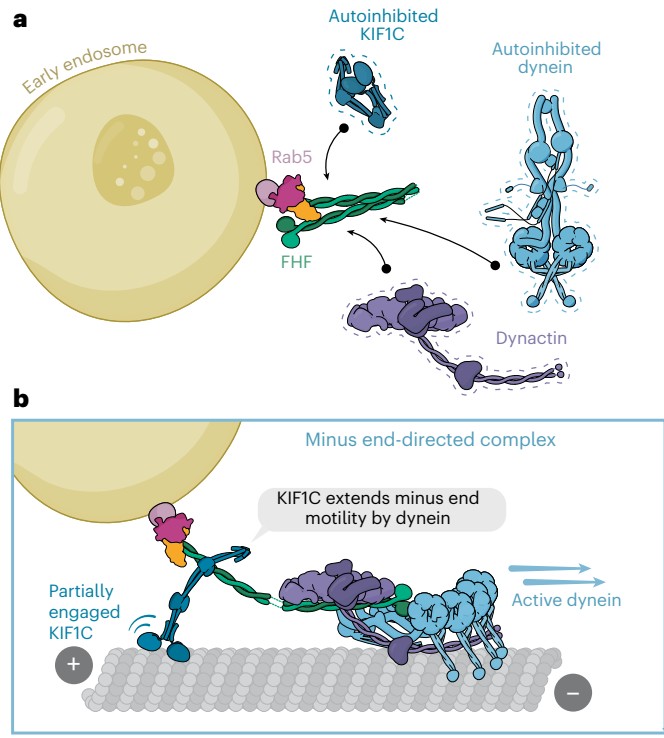

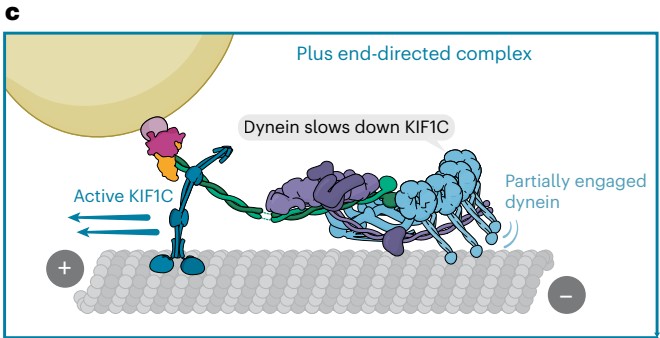

**Fig. 7 | Proposed model for activation of dynein by KIF1C. a**, FHF assembles on Rab5 marked early endosomes. The FHF complex is inherently autoinhibited through HOOK3 intramolecular interactions. **b**, KIF1C, which is autoinhibited, binds (through its stalk domain) to the HOOK3 C terminus and relieves the HOOK3 fold-back conformation. This enables dynein–dynactin access to the HOOK3 N terminus and minus end motility. During these runs, KIF1C only partially engages the microtubules, acting as a processivity factor for dynein. **c**, Dynein–dynactin, at sufficiently high local concentration, binds HOOK3 at its N terminus, exposing the C terminus for KIF1C binding and activation. Dynein weakly engages microtubules and slows down plus end transport.

to cellular cues and obstacles to trigger a directional switch in a timely manner. KIF1C-dependent cargoes, such as integrin-containing recycling endosomes and neuronal dense core vesicles, show directional reversals in cells[12,57], and we also observe them in our in vitro system, albeit at a low frequency. Reversals in cells could be more common due to the presence of mixed polarity microtubules and obstructions such as microtubule crossovers, ends, lattice defects and binding proteins. In addition to such mechanical effects, binding of regulatory factors and posttranslational modifications could trigger switch frequencies in a spatially controlled manner in cells.

### Linking motors through cargo adapters avoids a tug of war

A key feature of our reconstituted system is that motility of cocomplexes is predominantly unidirectional to either end of the microtubule at a velocity comparable to the unperturbed motors. Importantly, the nondriving motor remains bound to the cocomplex during runs and engages the microtubule but not strongly enough to stall the driving motor. This is in contrast to a number of previous studies with DNA-linked dynein and kinesin-1 or dynein and kinesin-3 (KIF1A) cocomplexes, which engage in a tug of war against each other and result in little overall movement[32–34]. One could argue the pauses observed in DDHK runs (Extended Data Fig. 9) might constitute phases in which the motors undertake tug of war. Although if this was the case, we would expect a higher occurrence of directional switches as symmetry breaking after a period of force balance should not consistently favor the direction before pause. Nonetheless, both dynein and kinesin linked by the adapter HOOK3 undertake far more processive runs than when motors are artificially linked.

Several other coordinated adapter-linked dynein–kinesin complexes have recently been described. One study revealed that the loading of kinesin-1 onto oskar ribonucleoprotein particles via tropomyosin1-I/C enables dynein–dynactin–BICD-mediated cotransport of the mRNA cargo into the oocyte. In contrast to our data, kinesin does not interfere with dynein motility due to tropomyosin1-I/C reinforcing a kinesin-1 autoinhibitory conformation. Only once dynein reaches the oocyte can the carried kinesin-1 be activated to continue mRNA transport to the posterior pole[58]. In contrast, a study using truncated TRAK2 to scaffold dynein and kinesin-1 showed that these cocomplexes moved solely in the plus end direction and the displacement of kinesin-1 was required for dynein-mediated movement to occur[27]. In yet another system, directional switching was observed when kinesin-3 KIF16B and dynein–dynactin–BICD2 complexes were tethered independently of each other to lipid vesicles. Modeling suggested that stochastic attachment and detachment of a reservoir of motors determined directionality and symmetry breaking after periods of tug of war[59]. In our adapter-linked system, instead, we show that directionality is an intrinsic property of the cocomplex and that diffusion of motors is fundamentally not required.

Why nondriving motors adopt a low resistance mode in our system, instead of fully engaging or fully detaching, is unknown. However, we have previously shown that KIF1C has an inherent tendency to slip backward under load rather than fully detach from the microtubule[60], consistent with its ability to extend minus end runs of DDH through microtubule tethering. The type of linkage may also play a role, as the kinesin-3 KIF1A, which has an even higher propensity for backslipping[61], nonetheless engaged in a tug of war when coupled to dynein using a DNA-based linker[34]. We propose that the reduced stiffness of the cargo adapter compared with DNA-linked systems or the ability to transmit conformational changes might enable the nondriving motor to switch into a conformation that weakly engages microtubules. Steric constraints through dynactin or additional adapter-mediated interactions that revert motors to their inhibited state might also enable the cooperation of KIF1C and dynein. Structural insights into the engagement of both motors with the adapter will make it possible to start to reveal how opposite polarity motors are coordinated during bidirectional transport.

## Online content

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

## Methods

### Cloning of protein expression constructs

Insect expression vector pFastBac-M13-8xHis-ZZ-LTLT-HOOK3-SNAPf (Addgene 130976) was made by amplifying HOOK3 from a human complementary DNA library by PCR using oligonucleotides AS689 (ACAT-AggcgcgccctATGTTCAGCGTAGAGTCGCTG), which encoded a 5′ AscI site and AS691 (ACAATaaccggtggatcCCTTGCTGTGGCCGGCTG), which introduced a 3′ AgeI/BamHI site. The PCR products were cut with AscI and AgeI/BamHI and ligated into pFastBac-M13-8xHis-ZZ-LTLT-BICD R1-SNAPf opened with AscI and BamHI/AgeI sites to replace BICDR1. This attached an N-terminal purification tag containing 8xHis, tandem protein-A binding motifs (ZZ) and tandem recognition sites for Tobacco Etch Virus peptidase (TEV) (LTLT) and a C-terminal SNAPf domain.

An expression vector for HOOK3 without a C-terminal tag was produced by PCR using AS689 and AS967 (ATAAATgcggccgcCGac-cggtttaCCTTGCTGTGGCCGGCTG). This was cut with AscI and NotI and then inserted into pFastBac-M13-8xHis-ZZ-LTLT-HOOK3-SNAPf, replacing HOOK3-SNAPf with HOOK3 followed by a stop codon and creating pFastBac-M13-8xHis-ZZ-LTLT-HOOK3 (Addgene 222289).

For all FHF plasmids used in this study, the three human FHF genes were first codon optimized for expression in Sf9 cells and synthesized commercially by Twist or Epoch Life Science. The genes were first subcloned into pACEBac1 vectors using Gibson cloning[62]. In the case of HOOK, the full-length or truncated form of the gene (for example encoding for HOOK3, HOOK2, $H^N$ or $H^c$) was subcloned into a pACE-Bac1 vector encoding a strep II tag followed by a PreScission protease cleavage site (added in frame at the 5′ end). A pBig1 plasmid for FHF coexpression was assembled using the biGBac system as described previously[42,63], where FTS, FHIP1B and HOOK pACEBac1 vectors were used as backbones. Correct assembly of FHF was verified by Sanger sequencing (Source Bioscience). Constitutively active HOOK3[1–522] was constructed as described previously[21]. All primers used for cloning these constructs are listed in Supplementary Table 1.

The GST–KIF1C stalk construct was cloned into a modified pGEX-6P2 vector (pGEX-6P2-LTLT), which contains an insertion of a tandem TEV cleavage site between the BamHI and EcoRI sites. Amino acids 642–922 from human KIF1C were amplified from pKIF1C-2xFlag using AS696 (GAGTCgaattcGAAATGGAGAAGAGGCTGCAG) and AS778 (ATAAATgcggccgcCTACTCCCAGCTTGACAGTGGTG), digested with EcoRI and NotI, and ligated into pGEX-6P2-LTLT to create pGEX-6P2-LTLT-KIF1C(642–922)-6His (Addgene 222289). The GST–KIF1C stalk–GFP (pET22b-GST-6P2-LTLT-KIF1C(642–922)-GFP-6His, Addgene 222288) bacterial expression plasmid was created by opening pET22b-HsCLASP2-EBBD-eGFP-6His between the NdeI and SalI sites and ligating in a PCR product of GST–KIF1C stalk (previously described) formed between oligonucleotides AS777 (TGCCAcatATGTCCCCTATAC-TAGGTTATT) and AS780 (ATATTgtcgaccCTCCCAGCTTGACAG).

All other variants of the GST–KIF1C stalk construct (KIF1CS[2]: 674–922, KIF1CS[3]: 674–822, KIF1CS SGD: 720–841, KIF1C stalk ΔSGD_1: 642–922 Δ722–840, KIF1C stalk ΔSGD_2: 642–922 Δ722–840) were based on the original GST–KIF1C stalk plasmid (Addgene 222288), where fragments were PCR amplified at the indicated sequence positions and assembled using Gibson cloning[62]. All primers used for cloning these constructs are listed in Supplementary Table 1.

### Expression of HOOK3

Sf9 insect cells (VWR, EM71104-3) were maintained in ExCell 420 serum-free media (Sigma, 14420C). For baculovirus expression of HOOK3, pFastBac-M13-derived plasmids were transformed into DH10BacYFP competent cells and plated on LB agar supplemented with 30 μg ml$^{-1}$ kanamycin (Sigma, K4000), 7 μg ml$^{-1}$ gentamycin (Sigma, G1372), 10 μg ml$^{-1}$ tetracycline (Sigma, T3258), 40 μg ml$^{-1}$ iso-propyl β-D-1-thiogalactopyranoside (Melford, MB1008) and 100 μg ml$^{-1}$ X-Gal (Melford, MB1001). Positive transformants (white colonies) were screened by PCR using M13 forward and reverse primers for the

integration into the viral genome. The bacmid DNA was isolated from the positive transformants by the alkaline lysis method and transfected into Sf9 cells with Escort IV (Sigm, L-3287) according to the manufacturer's protocols. After 5–7 days, the virus (passage 1, P1) is collected by centrifugation at 300g for 5 min in a swing out 5804 S-4-72 rotor (Eppendorf). The P1 virus was used to infect 50 ml of Sf9 culture and P2 virus was collected after 3–5 days through centrifugation and subsequent filtration through 0.45 μm polyvinylidene difluoride membrane (Merck Millipore, SLHV033RS). P2 virus was either propagated further or stored at 4 °C in the dark. For protein expression, Sf9 cells were infected at a density between 1.5 and 2 × 10$^6$ cells per milliliter with 1:100 dilution of P2 or P3 virus. The cultures were grown for 48–72 h or until all cells contained yellow fluorescent protein (YFP) fluorescence under the microscope. The cells were collected by centrifugation at 500g for 15 min, and the pellets were stored at −80 °C until use.

### Expression of FHF

All FHF constructs were expressed using the baculovirus-Sf9 system. Bacmid DNA was generated by transformation into DH10EmBacY cells (comprising a YFP reporter gene). The white colonies were picked from selective plates and the bacmid DNA were recovered using a modified alkaline lysis Qiagen mini-prep protocol as described previously[15]. To generate the P1 virus, 2–3 μg bacmid DNA was transfected (FuGene HD, Promega) into 2 ml Sf9 cells plated at 0.5 × 10$^6$ cells ml$^{-1}$ density in a six-well plate and left to incubate for 5–7 days at 27 °C without shaking. The success of transfection was monitored by the intensity of green fluorescence emitted from cells under a fluorescence microscope. These P1 viruses were collected by pipetting and amplified to create P2 virus: 1 ml P1 was added to 50 ml Sf9 cells at 0.5 × 10$^6$ cells ml$^{-1}$ in a 250 ml Erlenmeyer flask to give a 1:50 v/v dilution. This was incubated at 27 °C with 140 rpm shaking for 72 h. The P2 virus was collected (supernatant) by spinning at 4,000g for 5 min at room temperature. For protein expression, 5–7 ml of P2 virus was added to 500 ml of Sf9 cells at 1.5 × 10$^6$ cells ml$^{-1}$ in roller bottles and left to incubate at 27 °C with 140 rpm shaking for 56 h. At this time point, the cells were collected by centrifugation (4,000g for 10 min at 4 °C) and snap frozen in liquid nitrogen before storage at −80 °C.

### Purification of full-length HOOK3

Unless otherwise stated, all protein purification procedures were performed at 4 °C. Sf9 cells were lysed in lysis buffer (50 mM HEPES buffer pH 7.4, 150 mM NaCl and 20 mM imidazole) supplemented with 1× complete protease inhibitor cocktail (Roche, 6538282001). Lysis was achieved in a glass tissue douncer with the tight pestle using 20–30 strokes. The resultant lysate was cleared by centrifugation at 50,000g for 40 min. The cleared lysate was loaded on to 12 ml of NiNTA agarose beads (Qiagen, 30250) and batchbound for 1.5 h. The beads were transferred to a gravity flow column and washed with 100–200 column volumes (CVs) of lysis buffer followed by 100–200 CVs of wash buffer (50 mM HEPES pH 7.4, 150 mM NaCl and 60 mM imidazole). The protein was eluted in up to 10 ml of elution buffer (50 mM HEPES, pH 7.4, 150 mM NaCl and 300 mM imidazole), which was subsequently batch-bound to 1 ml immunoglobulin G (IgG) Sepharose (Cytiva, 17096901) for 1.5 h. The protein-bound IgG beads were washed with 100 CVs of TEV cleavage buffer (50 mM Tris buffer pH 7.4, 148 mM KAc, 2 mM MgAc, 1 mM EGTA buffer and 10% glycerol), transferred to a 1.5 ml tube and incubated with 35 μM benzylguanine conjugated fluorophore (NEB, S9136S) for 2 h before being returned to the column, washed with a further 100 CVs of TEV cleavage buffer and cut off the beads with 40 μg ml$^{-1}$ TEV protease at 25 °C for 1 h. The protein was collected from the beads and the bead bed was washed with a further 1 CV of buffer. The collected protein was subjected to a clearing spin at 19,000g at 4 °C for 5 min before being aliquoted and stored in N$_2$ vapor.

Untagged HOOK3 was expressed from pFastBac-M13 -8xHis-ZZ-LTLT-HOOK3 as described above and purified by a singlestep

IgG Sepharose purification followed by gel filtration over a Superose 6 increase 10/300 column. The purification buffer used throughout was 50 mM HEPES 7.5, 150 mM NaCl, 2.5 mM MgSO$_4$, 10% glycerol, 2 mM dithiothreitol (DTT), 0.1 mM ATP and 0.05% Triton X. Lysis was performed using 20–30 strokes in tissue douncer, cleared lysates were bound to 750 µl of IgG Sepharose beads for 3 h, and the beads were washed with 300–400 ml of purification buffer. The beads were resuspended in a 1.5 ml tube and incubated with 40 µg ml$^{-1}$ TEV protease for 1 h at 25 °C with agitation. The eluate was collected in a gravity flow column, the beads were washed with an additional 1 CV of buffer and then the protein was concentrated using a 30 kDa centrifugal concentrator to a final volume of around 600 µl. The protein was clear spun at 19,000g at 4 °C for 5 min before being injected into a pre-equilibrated Superose 6 increase 10/300 column in gel filtration buffer (50 mM HEPES 7.4, 150 mM NaCl, 2.5 mM MgSO$_4$ and 1 mM DTT). The peak corresponding to HOOK3 at around 11–12 ml was concentrated in the presence of 10% glycerol and then aliquoted, snap-frozen and stored in N$_2$ vapor.

### Purification of FHF complexes

A total of 2 l worth of pellets were thawed in a water bath and resuspended with 45 ml lysis buffer (50 mM HEPES pH 7.2, 150 mM NaCl, 10% glycerol and 1 mM DTT) supplemented with 2 mM phenylmethyl sulfonyl fluoride and two complete EDTA buffer-free protease inhibitor tablets (Roche, 6538282001) per 50 ml of buffer. The cells were lysed using a tight dounce homogenizer using 20–25 strokes on ice. The cell debris was pelleted by spinning lysates at 504,000g for 45 min at 4 °C in a Ti70 rotor in an ultracentrifuge (Beckman Coulter). In a cold room, the clarified lysate was applied to 4 ml pre-equilibrated Strep-Tactin Sepharose beads (IBA, 2-1201-025) in a Bio-Rad gravity flow column and the flow through was immediately collected. The flow through was reapplied to the beads to ensure full capture of protein. The beads were washed with ~450 ml lysis buffer followed by 200 ml PreScission (Psc) buffer (50 mM Tris–HCl pH 7.4, 150 mM NaCl, 1 mM EDTA and 1 mM DTT). The beads + 6 ml Psc buffer were collected with cut pipette tip and distributed equally across two 5 ml Eppendorf tubes. To cleave bound protein from beads, 90 µl Psc enzyme at 2 mg ml$^{-1}$ was added to each tube and left to incubate in a PTR-60 multirotator (Grant-bio) at orbital setting 2 for 16 h at 4 °C. The following day, the bead-elution mixture was added to a small gravity column and washed with 5 ml Psc buffer to release remaining loosely bound protein. The elution was concentrated in a 15 ml 100 kDa cutoff (4,000g, 5 min spins) and mixed in between spins by pipetting until reaching ~8 mg ml$^{-1}$ concentration in ~800 µl volume. The concentrated protein was spun down in a microfuge tube within a microcentrifuge (12,000g for 5 min). A total of 250 µl protein was injected into a Superose 6 10/300 column using a 500 µl loop, collecting 300 µl fractions. The peak fractions were pooled and concentrated in a 4 ml 100 kDa cutoff concentrator. A total of 10% glycerol was added, and the protein was aliquoted into 5 µl lots. The typical yields from these purifications ranged from 2 to 4 mg.

### Purification of KIF1C stalk constructs

All KIF1C stalk constructs used in this study, apart from GST–KIF1C stalk–GFP, were purified as follows. A total of 1–2 l worth of SoluBL21 pellets expressing the relevant GST–KIF1C stalk construct were resuspended to a homogeneous mixture with 50 ml lysis buffer (50 mM Tris pH 7.4, 300 mM NaCl, 20 mM imidazole, 10% glycerol and 1 mM DTT) supplemented with 2 mM phenylmethyl sulfonyl fluoride and one complete EDTA-free protease inhibitor tablet (Roche, 6538282001). The cells were lysed by sonication, and the lysate was cleared by spinning down in a Ti70 rotor at 433,706g (4 °C) for 30 min (Beckman Coulter). The cleared lysate was briefly incubated (approximately 5 min) with 3 ml Ni-NTA agarose (Qiagen, 30210) before the flow through was collected. The beads were washed with approximately 250 ml lysis buffer followed by 200 ml high salt lysis buffer (50 mM Tris pH 7.4,

500 mM NaCl, 30 mM imidazole, 10% glycerol and 1 mM DTT) and back to an additional 200 ml lysis buffer. The protein was eluted using elution buffer (50 mM Tris pH 7.4, 300 mM NaCl, 300 mM imidazole and 1 mM DTT) in 1 ml increments until 15 ml was collected. The combined elution was concentrated in a 30–100 kDa cutoff 15 ml concentrator (depending on the KIF1C stalk construct size) before injection into a Superdex 200 increase 10/300 column (Cytvia) pre-equilibrated in GF300 (25 mM HEPES 7.2, 300 mM NaCl, 1 mM MgCl$_2$ and 1 mM DTT). The peak fractions were pooled from multiple consecutive runs and concentrated down in a 30 kDa cutoff 4 ml concentrator until ~150–200 µl at 3 mg ml$^{-1}$. The fractions were snap frozen and stored at −80 °C until further use.

### Purification of other proteins

KIF1C–GFP and GST–KIF1C stalk–GFP constructs[23], dynactin[64], Tetramethylrhodamine (TMR)-labeled human dynein[15] and Lis1 (ref. [65]) were expressed and purified as described previously.

### TIRF assays of HOOK3-containing motor complexes

Menzel Gaser no. 1.5 22 mm × 22 mm and 22 mm × 50 mm coverslips were placed in holding racks and submerged in 6.4% w/v HCl at 60 °C for 16 h. The HCl was exchanged for distilled water and the coverslips were sonicated in a ultrasonic waterbath for 5 min at a time, exchanging the distilled water between sonications. The sonication step was repeated five times and then coverslips were dried with compressed air. The coverslips were stored between layers of lens tissue (Ross Optical, AG806) inside pipette boxes until they were used (maximally stored for a month). Before the imaging chamber assembly, the acid washed coverslips were plasma cleaned in a Henniker plasma clean (Henniker Plasma, HPT200) for 5 min. For each flow chamber, a 75 mm × 25 mm SuperFrost Plus glass slide (Thermo Scientific, J1800AMNZ) was taken and double-sided tape (Tesa, 64621) was laid in two parallel lines with a gap of 5 mm in between them. A single coverslip was laid over the tape and flattened to create an enclosed chamber with a volume of approximately 1,015 µl. In experiments where additional components would be added to the TIRF flowchamber while it was on the microscope, an open chamber design was adopted. The chambers were constructed between a 22 mm × 50 mm coverslip with a 22 mm × 22 mm coverslip on top. By imaging through the 22 mm × 50 mm coverslip, the top of the chamber was left open, allowing additional components to be flowed into the chamber in situ. These chambers were not as hydrophilic as those created with the SuperFrost Plus coated glass slides, and therefore, they were prewet with 0.1% v/v Triton X100 to prevent formation of air gaps during filling.

The microtubules were polymerized in a final volume of 20 µl MRB80 (80 mM PIPES buffer pH 6.8, 4 mM MgCl$_2$, 1 mM EGTA and 1 mM DTT) using 82 µg of unlabeled porcine tubulin with 2.5 µg of biotin-labeled porcine tubulin (Cytoskeleton, T333PB). The mixture of diluted tubulin was pipetted to mix and then spun for 5 min in a chilled airfuge (Beckman Coulter, 340401) at around 105,000g. The spun tubulin mixture was placed into a 0.5 ml tube, a final concentration of 5 mM GTP was added and the microtubules were allowed to polymerize for 1 h at 37 °C. After 1 h, the mixture was topped up to 100 µl by the addition of 80 µl MRB80 containing 50 µM paclitaxel (Alfa Aesar, J62734. MC), flicked to mix and kept at room temperature. The next day, the microtubules were pelleted in a benchtop centrifuge at 19,000g for 12 min, the supernatant containing any unpolymerized tubulin was removed and replaced with 100 µl MRB80 containing 50 µM paclitaxel. The microtubule pellet was flicked to resuspend it and once it had been broken up, the microtubules were pipetted gently 510 times with a 200 µl pipette tip until no traces of the pellet were remaining. The microtubules prepared in this way were kept for 34 weeks with the tube wrapped in aluminum foil and diluted 1:50 in taxol containing buffer before use in microscopy chambers. This timeframe allowed for microtubule concatenation, giving rise to long microtubule substrates

for transport assays, which are otherwise difficult to obtain through freshly assembled preparations.

Throughout the following steps, flow chambers were kept upturned in humidified chambers to avoid drying. The chambers were first coated with PLL(20)g[3.5]PEG(2)/PEG(3.4)biotin(50%) (Susos) by flowing in 12 μl of a 0.2 mg ml$^{-1}$ solution in MRB80. The chambers were incubated for at least 10 min before being washed with 30 μl TIRF assay buffer (TAB: 25 mM HEPES pH 7.2, 5 mM MgSO$_4$, 1 mM EGTA, 1 mM DTT and 10 μM paclitaxel) and then, 12 μl of 0.625 mg ml$^{-1}$ streptavidin (Merck, S4762) in TAB was added. Excess streptavidin was washed away with 30 μl TAB before 14 μl of 1:50 diluted polymerized GDPtaxol microtubules were added to the chamber. The microtubules were allowed to attach for 1,020 s before the unstuck microtubules were washed away with 14 μl TAB. The chamber was blocked for a period of 12 min using 1 mg ml$^{-1}$ κcasein (Merck, C0406) in TAB buffer. Excess κcasein was washed away with 14 μl TAB and then the reaction mixtures were flowed into the chamber.

The reaction mixtures were prepared in TAB supplemented with 0.2 mg ml$^{-1}$ κcasein, 25 mM KCl, 20 μM paclitaxel, an ATP regenerating system (5 mM ATP (Melford, B3003), 5 mM phosphocreatine (Merck, P7936) and 7 U ml$^{-1}$ creatine phosphokinase (Merck, C3755)) as well as an oxygen scavenger system (0.2 mg ml$^{-1}$ catalase (Merck, C9322), 0.4 mg ml$^{-1}$ glucose oxidase (Merck, G7141), 4 mM DTT and 50 mM glucose). A total of 200 nM TMR–dynein, 200 nM dynactin, 1,600 nM labeled adapters and 180 nM KIF1CGFP were mixed in 2.5 μl of dynein storage buffer GF150 (25 mM HEPES pH 7.4 and 150 mM KCl) and then diluted with 2.5 μl TAB buffer, which contains no added salt and thus the final salt concentration during complex assembly was ~75 mM KCl. As the concentration of unlabeled HOOK3 was lower than that of tagged cargo adapters, an excess of Lis1 was added to aid complex formation. The complexes with unlabeled HOOK3 were formed by mixing 50 nM of each dynein, dynactin and KIF1CGFP or KIF1C buffer control in the presence of 250 nM unlabeled HOOK3 and 1250 nM unlabeled Lis1. The complex mixture was pipetted, spun down and then retained on ice for 45–60 min. Just before use, the complexes were mixed once more by pipetting before being diluted 1:20 or 1:40 into the reaction mix. The reaction mix was added to the flow chamber and imaging started immediately using an Olympus TIRF system with a ×100 numerical aperture 1.49 objective, 488, 561 and 640 nm laser lines, an ImageEM electron multiplying charge coupled device camera (Hamamatsu Photonics) under the control of xCellence software (Olympus) and an environmental chamber maintained at 25 °C (Okolab). Emission resulting from illumination with 488 or 561 nm laser lines was filtered by bandpass filters controlled by a filter wheel (Olympus) to avoid bleed through.

## Motility analysis of HOOK3-containing motor complexes

The microtubules were manually traced in a maximum intensity projection or reference image and then the kymographs were generated for all the channels that were recorded using a custom ImageJ macro. Regions of interest were saved to return to the microtubule path if required. Next, motor tracks were manually traced with multisegmented lines in kymographs and the phase durations, lengths and speed were stored in a .csv file, while the line was also saved as a regions of interest file. A Python analysis software package reads the saved tracks and classifies and summarizes the motility parameters (run length, dwell times, run speeds, landing rates and directionality) at the single-motor level, the permicrotubule level, the permovie level and the perexperiment level. To do this, the paths were segmented into runs. A run was defined as the distance covered before falling static (having an absolute speed of less than 25 nm s$^{-1}$) or the distance covered before changing direction (having a speed of more than 25 nm s$^{-1}$ in one direction and then a speed of more than 25 nm s$^{-1}$ in the opposite direction). For each segment of the path, it was determined whether this segment was going toward the plus or minus end of the microtubule (or whether the motor was paused) and the time spent in each mode of transport was tallied. The

total time a motor spend on the microtubule was its dwell time. The total run length in the plus or minus end direction was also tallied for the individual runs within the track. The total run length was defined as the sum of the absolute values of all individual runs for one motor. The average speed of a particle was defined as the total run length divided by the total dwell time. The landing rate was calculated as the number of tracks per kymograph width (in nm) and length (in s) and converted to μm/min by multiplying with 60,000. The overall directionality of a track was defined in the following way. First, any track with a total run length of less than 1,000 nm was classified as 'static'. Any remaining track that traveled only toward the plus end was classified as 'plus end directed', while any track that traveled only toward the minus end was classified as 'minus end directed'. Any track that traveled more than 1,500 nm in the plus and minus end directions was classified as 'bidirectional'. After this classification, any remaining track that traveled more toward the plus end than the minus end was labeled 'plus end directed', while the opposite was labeled 'minus end directed'. Therefore, a track that traveled 20,000 nm in the minus end direction and 200 nm in the plus end direction would still be classified as 'minus end directed'.

## Statistical analysis of motility of HOOK3-containing complexes

Statistical analyses were performed in Python with use of the following modules: scipy[66] and scikit-post hocs. All statistical analyses were performed as follows. The data were tested to see whether they followed a normal distribution using D'Agostino and Pearson's test. If data were normally distributed, pairwise interactions were tested using a two-tailed $t$-test for independent samples, and the resulting $P$ values were corrected for multiple comparisons if necessary. Where $t$-tests have been used, this is indicated in the figure legend; otherwise, all statistical analyses relate to the following nonparametric testing process. If one or more experimental groups were not normally distributed, a Kruskal–Wallis $H$ test was used to determine whether any of the medians of the experimental groups differed. If the Kruskal–Wallis $H$ test showed that one group was significantly different, then pairwise interactions were tested using Conover's post hoc test. The $P$ values of these pairwise interactions were corrected for multiple comparisons.

The data were plotted using matplotlib[67] and the superplots were created with a custom-made extension to matplotlib. The graphs were edited in Adobe Illustrator (Adobe), where modifications were limited to colors, line widths, spacing and text size.

Microscopy images were prepared in FiJi[68]. The image manipulations were limited to scaling of contrast (between minimum and maximum values) and displaying images with different color lookup tables. Where images from the same experiment are displayed alongside one another, their contrast is scaled equally.

## Size exclusion chromatography reconstitution of complexes

All gel filtration reconstitution experiments were performed using a Superose 6 Increase 3.2/300 column (Cytvia) using a 100 μl loop and 50 μl loaded sample in GF150 (GF150, 25 mM HEPES pH 7.2, 150 mM KCl, 1 mM MgCl$_2$ and 1 mM DTT). This column was either connected to an AKTAmicro or an AKTApure micro system (Cytvia). A typical reconstitution used 5 μM FHF construct and 10 μM KIF1C stalk construct diluted in GF150.

## Pulldown assays

All steps were performed at room temperature. A total of 200 μl Strep-Tactin Sepharose (IBA) slurry was added to disposable Poly-Prep chromatography columns (Bio-Rad) for each experimental condition. The bead storage solution was allowed to flow through followed by a 2 ml wash with GF50 buffer (GF150, 25 mM HEPES pH 7.2, 50 mM KCl, 1 mM MgCl$_2$ and 1 mM DTT). The samples comprised 5 μM strep-tagged bait (FH$^C$F) and 10 μM H$^N$ and/or 5 μM GST-tagged KIF1C stalk construct ± SGD. All samples were diluted with GF50 to bring to 60 μl and

then incubated for 15 min. The 10 µl sample input was taken away, and 50 µl samples loaded onto the beads to let bind for 15 min. The beads were washed twice with 1 ml of GF50. A total of 500 µl GF50 was then added, and the beads were resuspended. Then, 400 µl of this wash + bead sample was transferred to fresh microcentrifuge tubes. Beads were spun down 1,000g for 1 min and 250 µl of the supernatant was removed. To the remaining bead + buffer mixture (~150 µl volume), 30 µl of NuPAGE LDS Sample Buffer (4X) (Invitrogen) was added, and 30 µl of this bead sample was loaded into a ten-well NuPAGE 4–12%, Bis–Tris, 1.5 mm, Mini Protein Gel and ran using MOPS buffer.

## Mass photometry
To assess the oligomeric state of FHF with and without KIF1C stalk, we used a mass photometry machine (TwoMP version, Refeyn Ltd). Samples of ~10 µl were added to wells within gaskets attached to clean coverslips. These samples were diluted to 100 nM in GF150 buffer. The movies were imaged for 60 s using the AcquireMP software (Refeyn) and the results were processed using the DiscoverMP software. Gaussian fitting was used to determine the mass peaks against a calibration performed using similar buffer conditions.

## Cryo-EM sample preparation
A total of 4 µl of FHF at 0.9 mg ml$^{-1}$ containing either 0.1% CHAPSO or 0.005% Igepal was added to an unsupported Quantifoil (2/2 Au 300-mesh) grid and left to incubate for 15–20 s inside a Vitrobot mark IV before blotting for 1 s (−13 blot force) and plunge freezing into liquid ethane. The grids were stored in liquid nitrogen and screened for ice thickness and protein concentration using a Thermo Fisher Glacios transmission electron microscope equipped with a Falcon III (Thermo Fisher).

## Cryo-EM data collection
Electron micrograph movies were collected using a Titan Krios (300 keV X-FEG) (Thermo Fisher Scientific) equipped with a K3 direct electron detector and a BioQuantum energy-filter (datasets 1 and 3) (Gatan) or Falcon 4i pre-GIF direct electron detector (dataset 2) (Thermo Fisher Scientific). The movies were collected at 75,000× magnification in energy-filtered transmission electron microscopy mode using EPU 2.6.1 software, yielding a pixel size of 1.09 Å for datasets 1 and 3 (Fringe-Free Illumination) and 1.08 Å for dataset 2. Aberration-free image shift collection was used for data collection (five acquisitions per hole for K3 or four acquisitions per hole for Falcon 4). See Table 1 for acquisition parameter details.

## Cryo-EM image processing
All cryo-EM image processing steps were performed in RELION-4.1 (ref. 69) unless otherwise stated. The movies were motion corrected using the RELION implementation. The particles were picked in crYOLO using the general model trained on low-pass filtered images[70]. These particles were extracted (bin 2, giving a pixel size of 2.16–2.18 Å per pixel depending on the camera). At this point, datasets 1 and 2 were combined while dataset 3 was kept separate until later (Extended Data Fig. 2b). For datasets 1 and 2, a total of 2,507,930 particles were subjected to unmasked three-dimensional (3D) classification (five classes) with alignment using an ab initio generated initial model of FHF. The best class was selected (1,012,409 particles), and a global 3D refinement was performed, yielding an overall 4.8 Å resolution structure. This was Bayesian polished and rerefined, leading to an improved 4.5 Å resolution structure. At this point, particles were re-extracted to their unbinned parameters and subjected to a round of masked 3D classification without alignment. This gave two good classes, which were selected (781,492) for subsequent merging with dataset 3 particles.

Dataset 3 was processed as follows. A total of 2,368,723 extracted particles were used for cleaning steps using two-dimensional (2D) classification. A total of 1,232,461 particles were taken forward for unmasked

3D classification with alignment and the best class selected (723,461 particles) was subjected to 3D refinement (using global search parameters), giving a 4.4 Å structure. Dataset 3 particles were then re-extracted to their unbinned parameters (256 box size, 1.09 Å per pixel). The unbinned particles were used for a local 3D refinement, resulting in a 3.7 Å structure. Bayesian polishing was performed using this unbinned, consensus refinement, which improved the subsequent 3D refinement to 3.3 Å resolution. However, the structure was anisotropic at this point. To reduce anisotropy in the map, a combination of StarParser ([https://github.com/sami-chaaban/starparser](https://github.com/sami-chaaban/starparser)) and RELION's subset select tool was used to remove overrepresented views. StarParser was used to plot a histogram of the rlnAngleRot values from the 3D refinement particles. star file. On the basis of this visualization, the rlnAngleRot values were split into different bins using RELION's subset selection (metadata label: rlnAnglRot), for example, angles that fit in within −140 to 100 and between 100 and 200. This generated two particles .star files. The latter star file (100–200 rlnAngleRot) had the majority of overrepresented views, and this was subjected to random extraction of ~50% of particles using StarParser to reduce the overrepresentation. The output star file was then combined with the unmodified first star file (−140 to 100 rlnAngleRot) (JoinStar files in RELION), and duplicate particles were removed (the Subset selection tool in RELION).

At this point, datasets 1 and 2 and the trimmed dataset 3 particles were merged and a new consensus refinement was obtained of the merged dataset (total particles: 1,207,965) at 3.4 Å resolution. These particles were subjected to 3D classification without alignment (eight classes, T = 8, with a mask around the FHF (eight pixels binary map extension and eight pixels soft edge)). The class with the highest resolution features and most complete density was selected, giving 377,595 particles. As the overall structure still showed anisotropy at this point, the reduce anisotropy procedure described above was re-employed at this point. A total of 273,204 particles were subsequently taken forward to 3D refinement, contrast transfer function refinement (defocus and astigmatism), 3D refinement, contrast transfer function refinement (magnification anisotropy) before a final 3D refinement. After post-processing, this final map was at 3.2 Å resolution (after correction of calibrated pixel size at 1.059 Å per pixel) and was used for all subsequent model building and real-space refinement.

## Model building and refinement
The initial coordinates for docking into the FHF cryo-EM density were predicted using a local cluster installation of Colabfold 1.5.0 (ref. 71) (AlphaFold2[72]), where full-length sequences of human FHIP1B and FTS were provided alongside a dimer of truncated HOOK3 (571–718). This resulted in a good overall fit but required dividing up of FHIP1B into two halves to better fit the N termini and C termini of the alpha solenoid. In addition, a switch had to be made to place the correct loop–CTE helix–loop region within the HOOK3 density (the build of the second CTE helix was omitted from the model as this region was too flexible and no density could be observed). Coot[73] was also used for real-space refinement of backbone and sidechains within the density. The side chains were removed from low resolution density regions using Phenix[74] (version 1.14). The model was then real-space refined using Phenix (version 1.20) after adding hydrogens (using a reference_coordinate_restraints.sigma value of 0.0005). The hydrogens were removed, and the resulting model was inspected in Coot, where outliers were fixed before repeating the real-space refinement process one last time. All structural figures were prepared using UCSF ChimeraX[75].

## AlphaFold predictions
AlphaFold structures of FHF and FHF bound to kinesin stalk were predicted using a local cluster installation of Colabfold 1.5.0 (ref. 71), where homology searches were performed using MMseqs2 (ref. 76) and AlphaFold2-Multimer (v3) was utilized[72,77]. AlphaScreen ([https://github.com/sami-chaaban/alphascreen](https://github.com/sami-chaaban/alphascreen)) was used to screen pairwise

interactions of sequential dimeric KIF1C fragments (in 120-residue increments) against a dimer of HOOK3 encompassing CC3 and CC4. The 'alphascreen --show_top 0.7' command was used to rank and select the model with the best interaction-site predicted aligned error (PAE) for each prediction, where only those with scaled PAEs higher than the 0.7 threshold are output into a pdf file for shortlisting. General guidelines for interpreting AlphaFold2 predictions were followed in accordance with a recent perspective[78].

### Analysis of processive events in FHF-containing samples

Microscope coverslips (epredia 22 mm × 22 mm no. 1.5) were cleaned by loading onto ceramic holders, soaking in ~300 ml 3 M KOH in a 1 l beaker and sonicating for 30 min in a water bath sonicator. The coverslips were washed by transferring to a new beaker containing ~500 ml Milli-Q H$_2$O and washed thoroughly five more times using the same volume of water. The coverslips were sonicated in Milli-Q H$_2$O for 30 min and transferred to new beaker containing 96% ethanol. Ethanol-immersed coverslips were sonicated for 30 min and stored in fresh ethanol. Before use, the ethanol was removed from the coverslips using compressed nitrogen and placed in their ceramic holder in a plasma cleaner (Fischione instruments, 1070 NanoClean) and evacuated until 'High Vac' was reached. The coverslips were exposed to plasma for 3 min with a setup of 75% argon and 25% oxygen at 100% power.

Motility chambers were prepared as previously described[79]. Briefly, two strips of double-sided tape were applied to a glass slide (epredia ground 90° frosted, BS7011/2 1.0–1.2 mm) at ~5 mm apart. A precleaned coverslip was placed on top and firmly sealed to form a channel that can take 10–15 µl of liquid.

The microtubules were prepared by adding 6 µl of 13 mg ml$^{-1}$ unlabeled pig tubulin, 2 µl of 2 mg ml$^{-1}$ Alexa Fluor 647 tubulin, 2 µl of 2 mg ml$^{-1}$ biotin-tubulin in BRB80 buffer (80 mM PIPES pH 6.8, 1 mM MgCl$_2$, 1 mM EGTA and 1 mM DTT) and 10 µl polymix buffer (2× BRB80, 20% v/v dimethylsulfoxide and 2 mM MgGTP). This mixture was left to incubate for 2 h at 37 °C in a heat block filled with water. The microtubule mixture was diluted to 100 µl with BRB80-T (BRB80 supplemented with 10 µM Taxol) and spun down at 21,300g for 8.5 min at room temperature to remove excess tubulin. The pellet was washed by resuspending with 100 µl BRB80-T and flicking the bottom of the tube, then washed once more in this way. The pellet was resuspended with 80 µl BRB80-T, covered in foil and stored overnight before use. Before use, the microtubule pellet was resuspended by flicking and its concentration was adjusted using BRB80-T based on visualizing the density of filaments inside the motility chambers.

To prepare the TIRF channels, 10 µl pluronic-F127 was added to the motility chambers for 1 min before addition of 15 µl of 0.4 mg ml$^{-1}$ biotinylated poly(L-lysine)-g-poly(ethylene-glycol) (PLL-PEG-biotin) (SuSoS AG). The channel was washed with dynein lysis buffer (DLB) buffer (30 mM HEPES–KOH pH 7.2, 5 mM MgSO$_4$, 1 mM EGTA, and 1 mM DTT) before adding 15 µl of 1 mg ml$^{-1}$ streptavidin (NEB) and immediately washing with DLB-C-T buffer (DLB supplemented with 1 mg ml$^{-1}$ α-casein, 50 mM KCl and 5 µM Taxol). A dilution was made of the microtubule stock in BRB80-T (between 1 in 3 and 1 in 5 dilution), and 15 µl of this was flowed into the chamber followed by a 10 µl DLB-C-T wash. The stock complex was prepared in a 5 µl volume (diluted with GF150: 25 mM HEPES pH 7.2, 150 mM KCl, 1 mM MgCl$_2$ and 1 mM DTT) such that the molar ratios were as follows: labeled TMR(SNAPf)–dynein (100 nM):dynactin (100 nM):FHF (2 µM):Lis1 (6 µM):KIF1C stalk (1.2 µM). After 5 min incubation on ice, the complex was diluted to 1 in 5 in buffer DLB-C-T and a further 1 in 20 in the final reaction mixture (1 µl diluted complex, 15 µl DLB-C-T, 1 µl 20 mM MgATP, 1 µl catalase–glucose oxidase (0.2 mg ml$^{-1}$ catalase (Calbiochem) mixed with 1.5 mg ml$^{-1}$ glucose oxidase (Sigma-Aldrich)), 1 µl 9% (w/v) glucose, 1 µl 25% (v/v) 2-mercaptoethanol). A total of 15 µl of this reaction mixture was added to the TIRF motility chamber immediately before data collection.

TIRF movies of moving molecules and snapshots of associated microtubules were collected at room temperature on a Nikon Eclipse Ti inverted microscope with a Nikon 100× TIRF 1.49 NA 100× oil immersion objective equipped with a back illuminated emCCD camera (iXonEM+ DU-897E). µManager software was used to collect 500 frame movies at 100 ms exposure and 105 nm per pixel using a 561 nm (100 mW, Coherent Cube) laser or a 100 ms 1 frame snapshot using a 641 nm (100 mW, Coherent Cube) laser for the microtubules.

The data analysis was manually performed using FiJi[68]. Here, Z projections of tif stacks were used to draw segmented lines along individual microtubule tracks and the reslice tool was used to make kymographs for each microtubule. Processive events for single molecules were defined as complexes that associated with microtubules for ≥1.2 s and ≥525 nm (that is, at least five pixels on the y axis and five pixels on the x axis), based on previously published criteria[15]. The velocity and run length were calculated for each processive event using a 105 nm pixel size and 0.136 s interframe rate.

The statistical analysis was performed in GraphPad Prism version 9.0.0. All the individual data was plotted in column format (15 microtubules per n, n = 8, therefore, 120 microtubules per sample apart from the dynein–dynactin condition, which had 15 microtubules per n, n = 4, therefore, 60 microtubules). The data were tested to see whether they followed a normal distribution using D'Agostino–Pearson omnibus normality test, which indicated that the different experimental groups were not normally distributed. A Kruskal–Wallis H test corrected for multiple comparison (Dunn's) was then used to test statistical significance (applied to total n within each experimental group).

### Crosslinking mass spectrometry

Protein crosslinking reactions were carried out at room temperature for 60 min with 50 mg of complex present in 10 mM of 1-ethyl-3-(3-dimethylaminopropyl)carbodiimide (EDC) or 5 mM of disuccinimidyl sulfoxide (DSSO). The crosslinked protein was quenched with the addition of Tris buffer to a final concentration of 50 mM. The quenched solution was reduced with 5 mM DTT and alkylated with 20 mM idoacetamide. An established SP3 protocol[80,81] was used to clean up and buffer exchange the reduced and alkylated protein, shortly, and the proteins were washed with ethanol using magnetic beads for protein capture and binding. The proteins were resuspended in 100 mM NH$_4$HCO$_3$ and were digested with trypsin (Promega) at an enzyme-to-substrate ratio of 1:25 and protease max 0.1% (Promega). Digestion was carried out overnight at 37 °C. The clean up of peptide digests was carried out with HyperSep SpinTip P-20 (Thermo Scientific) C18 columns, using 80% acetonitrile as the elution solvent. The peptides were then evaporated to dryness via a Speed Vac. The dried peptides were suspended in 3% acetonitrile and 0.1% formic acid and analyzed by nano-scale capillary liquid chromatography with tandem mass spectrometry using an Ultimate U3000 HPLC (Thermo Scientific) to deliver a flow of 300 nl min$^{-1}$. The peptides were trapped on a C18 Acclaim PepMap100 5 µm, 100 µm × 20 mm nanoViper (Thermo Scientific) before separation on PepMap RSLC C18, 2 µm, 100 Å, 75 µm × 50 cm EasySpray column (Thermo Scientific). The peptides were eluted on a 90 min gradient with acetonitrile and interfaced via an EasySpray ionization source to a quadrupole Orbitrap mass spectrometer (Q-Exactive HFX, Thermo Scientific). The mass spectrometry data were acquired in data-dependent mode with a Top-25 method, high resolution scans and full mass scans were carried out (R = 120,000, m/z of 350–1,750), followed by higher energy collision dissociation with stepped collision energy range of 21%, 27% and 33% normalized collision energy. The tandem mass spectra were recorded (R = 30,000, AGC target of 5 × 10$^4$, maximum IT of 150 ms, isolation window m/z of 1.6, dynamic exclusion 50 s). For the crosslinking data analysis, the Xcalibur raw files were converted to MGF files using ProteoWizard[82], and the crosslinks were analyzed by MeroX[83]. Searches were performed against a database containing

known proteins within the complex to minimize analysis time with a decoy database based on peptide sequence shuffling and/or reversing. The search conditions used three maximum missed cleavages with a minimum peptide length of 5; crosslinking targeted residues were K, S, T and Y; and crosslinking modification masses were 54.01056 Da and 85.98264 Da. Variable modifications were carbmidomethylation of cysteine (57.02146 Da) and methionine oxidation (15.99491 Da). The false discovery rate was set to 1%, and the assigned crosslinked spectra were manually inspected.

### Reporting summary

Further information on research design is available in the Nature Portfolio Reporting Summary linked to this article.

## Data availability

Cryo-EM maps and atomic coordinates have been deposited in the Protein Data Bank and Electron Microscopy Data Bank, under the accession codes EMD-18302 and Protein Data Bank 8QAT for FHF structure and EMD-18303 for the FHF + KIF1C stalk. All gel filtration, SDS–polyacrylamide gel electrophoresis (PAGE), single-molecule microscopy, AlphaFold2, mass spectrometry and associated analysis tables have been deposited and are available via Zenodo at https://doi.org/10.5281/zenodo.10949991 and https://doi.org/10.5281/zenodo.11360634 (refs. 84,85). The plasmids generated in this study are available from Addgene (accession numbers 222287, 222288, 222289, 222299, 222300, 222301 and 222302). Source data are provided with this paper.

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

## Acknowledgements

We thank S. Chaaban (MRC Laboratory of Molecular Biology (LMB)) for setting up AlphaFold2 installations, writing and advising on AlphaScreen and StarParser code use and advice for structural predictions. We thank C. K. Y. Lau and K. Singh (MRC LMB) for AlphaFold2 analysis and model building assistance. We thank the MRC Laboratory of Molecular Biology Electron Microscopy Facility for EM data collection time and J. Grimmett, T. Darling and I. Clayson for scientific computing support. We thank S. McLaughlin and C. M. Johnson from MRC Laboratory of Molecular Biology Biophysics facility for assistance, implementation and analysis of size-exclusion chromatography with multi-angle light scattering, fluorescent polarization and mass photometry experiments. We thank J. Shi (MRC LMB) for providing and maintaining insect cell cultures for protein expression. We thank D. Roth (Warwick) for cloning pFastBac-M13-8xHis-ZZ-LTLT-HOOK3 and technical assistance. We are grateful to the Warwick Computing and Advanced Microscopy Unit for their support and assistance. This work was supported by a Sir Henry Wellcome Postdoctoral Fellowship to F.A.A. (218653/Z/19/Z), Wellcome Investigator Awards to A.P.C (210711/Z/18/Z) and A.S. (200870/Z/16/Z and 224563/Z/21/Z), Medical Research Council funding to A.P.C (MC_UP_A025_1011) and MRC PhD studentship to A.J.Z. (MR/N014294/1). The funders had no role in study design, data collection and analysis, decision to publish or preparation of the manuscript.

## Author contributions

F.A.A. performed FHF-containing sample protein purification, reconstitution assays, EM data analysis, structure determination,

single-molecule experiments and, with A.P.C. and R.F.W., cloned constructs. A.J.Z. performed the HOOK3-containing sample purifications, cloning, reconstitutions, single-molecule experiments and data analysis and, with C.E.S., purified dynein and dynactin. T.E.M., with F.A.A., performed the crosslinking mass spectrometry experiments. A.S. and A.P.C. guided the project. F.A.A., A.S. and A.P.C. prepared the paper with input from all the authors.

## Competing interests

The authors declare no competing interests.

## Additional information

**Extended data** is available for this paper at https://doi.org/10.1038/s41594-024-01418-z.

**Correspondence and requests for materials** should be addressed to Andrew P. Carter or Anne Straube.

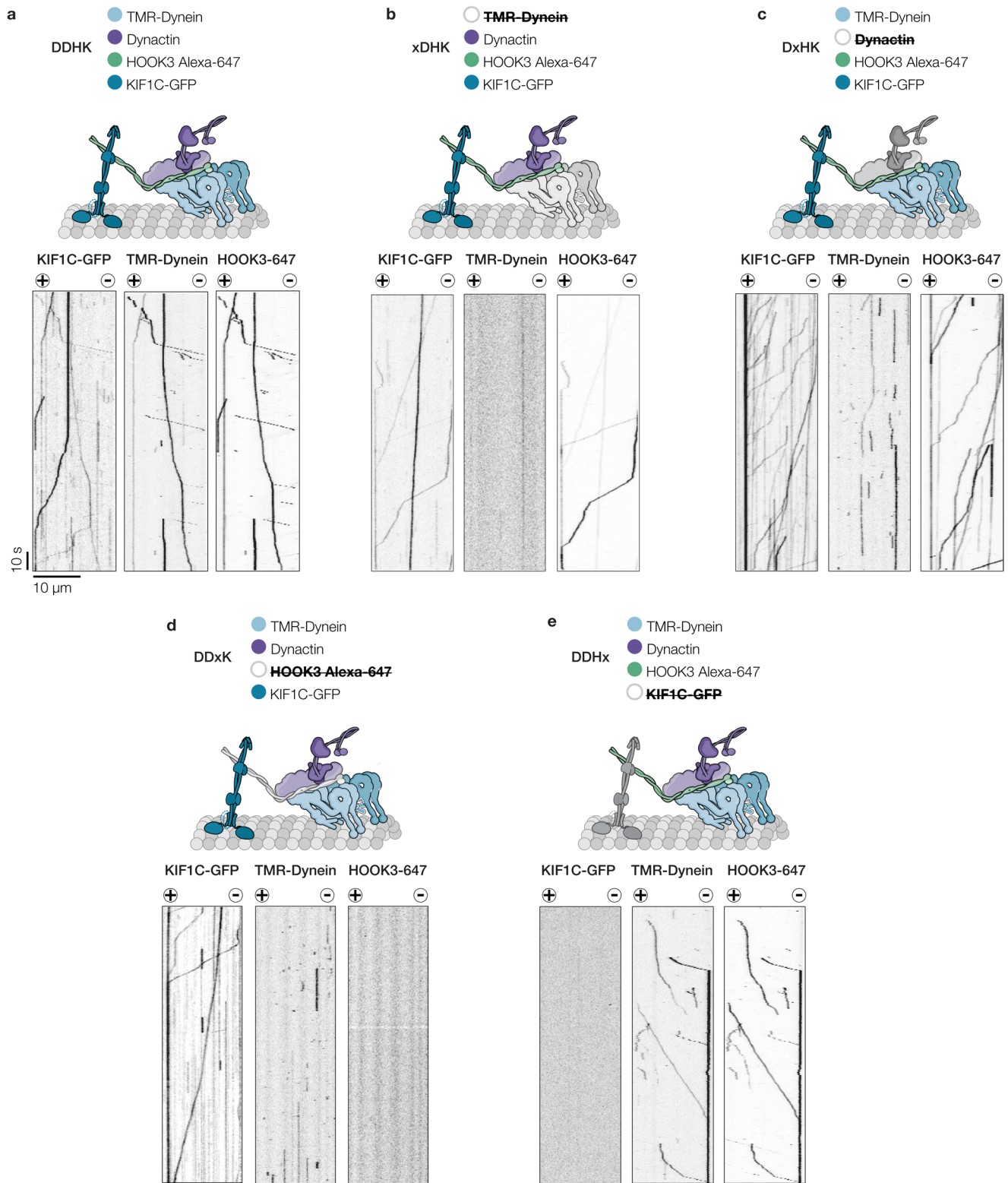

**Extended Data Fig. 1 | Controls for single molecule experiments.**
Representative kymographs from single molecule motility assays of Dynein-Dynactin-HOOK3-KIF1C (DDHK) complexes showing **a**, the full DDHK complex, and omission of individual components with **b**, TMR-dynein omitted (xDHK) and **c**, dynactin (DxHK) and **d**, HOOK3-Alexa 647 (DDxK) and **e**, KIF1C-GFP

(DDHx). The omitted factor is shown in grey in the cartoons and crossed out in the component list. Microtubule polarity is indicated with (+) and (−) signs. Note that motility of KIF1C and Dynein both towards the plus and minus end of microtubules is only observed in the complete DDHK complex.

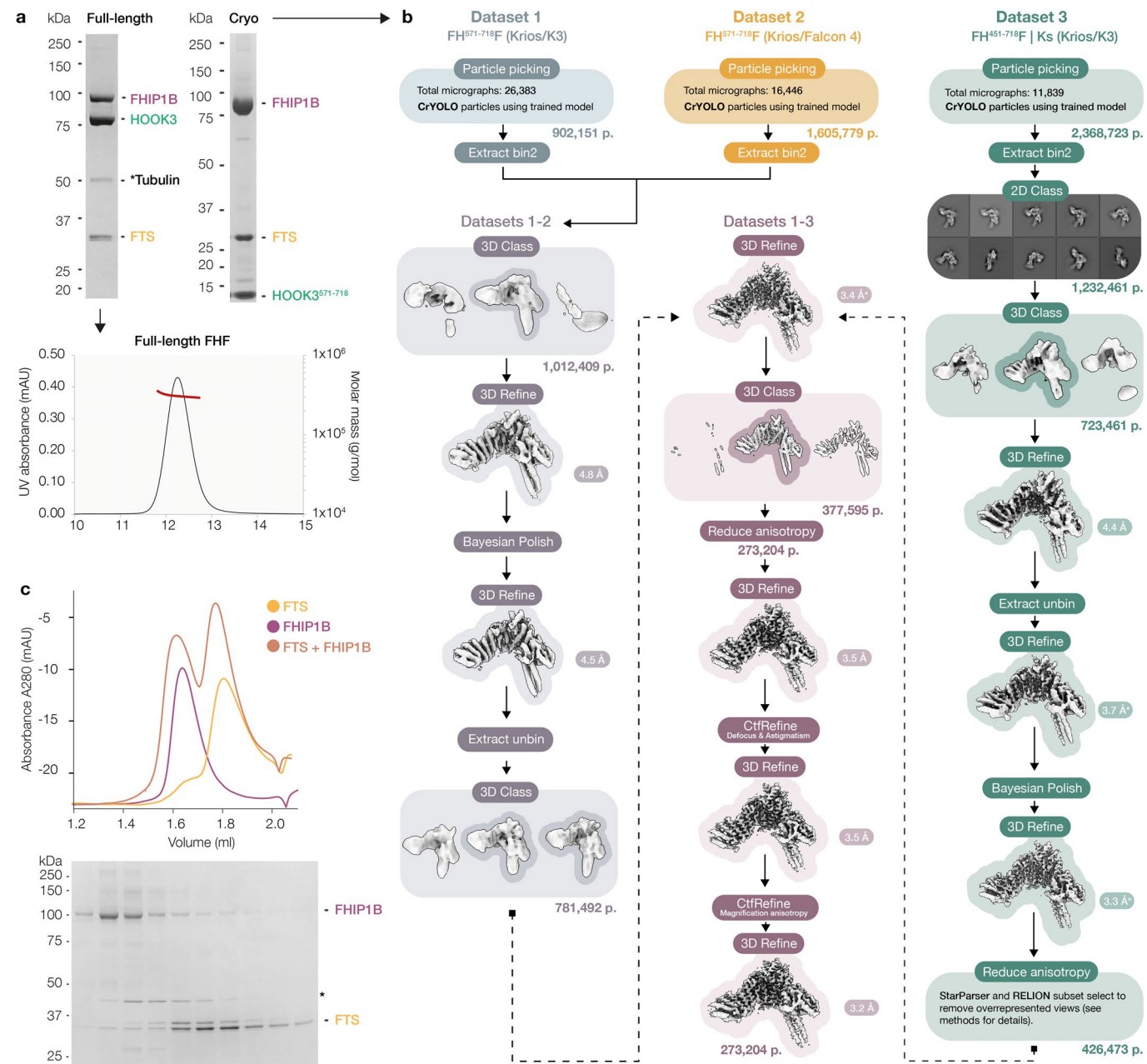

**Extended Data Fig. 2 | Biochemical analysis of FHF complex formation and cryo-EM processing workflow. a**, SDS-PAGE of the full-length (left) FHF complex and corresponding SEC-MALS chromatogram (below gel), revealing an expected molecular weight of ~304 kDa. On the right is one of the truncated FHF constructs used for cryo-EM structure determination (containing HOOK3 residues 571-718). Purification repeated three times for each. **b**, Cryo-EM processing workflow for three combined FHF datasets. All steps were performed in RELION-4.1[69] apart from particle picking, which was performed in CrYOLO[70]. Note that dataset 3 contained a construct of KIF1C stalk (GST-KIF1C stalk 674-922) but this did not alter the overall conformation of FHF as confirmed by comparison to FHF only datasets. Resolution values refer to post-sharpened and masked maps using the PostProcessing step of RELION. Asterisks are indicated after some resolution values to show that the number is a guide only due to map anisotropy leading to an overinflated estimate. **c**, SEC chromatogram of isolated 16 μM FTS, isolated 8 μM FHIP1B and 16 μM FTS + 8 μM FHIP1B added together. SDS-PAGE gel shows bands relevant to the combined FTS + FHIP1B run. Asterisk indicates unidentified contaminant. Experiment performed once.

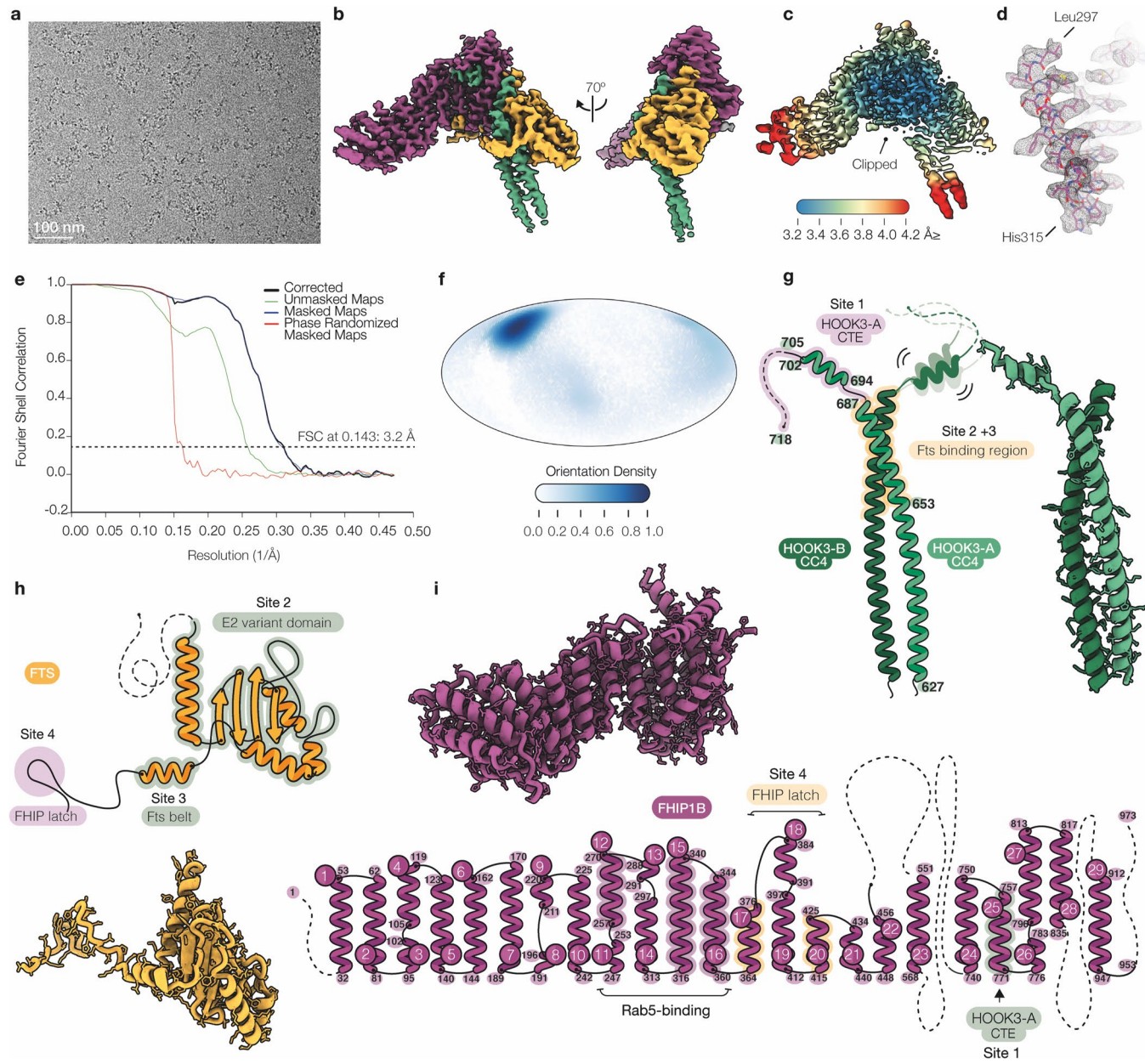

**Extended Data Fig. 3 | Validation and structural detail of FHF cryo-EM structure. a**, Integrated and motion corrected micrograph of FHF complexes from movie collected using a K3 camera in counting mode (1.09 Å/pixel). Data collection repeated three times with either isolated FHF (two datasets) or bound to KIF1C stalk (one dataset, where KIF1C stalk did not alter the FHF density). **b**, Segmented cryo-EM density of FHF complex shown in front and side views. **c**, RELION local-resolution plot applied to a clipped front view of the FHF complex. **d**, Example cryo-EM density (shown in mesh) from FHIP1B alpha helix (297-315) and corresponding built model. **e**, Gold standard Fourier shell correlation (FSC) curves as determined by RELION-4.1 (FSC = 0.143). **f**, 2D representation of angular distribution extracted from particles.star of the final structure. **g-i**, Topology and secondary structure of g, HOOK3 h, FTS and i, FHIP1B. Interacting regions with other FHF components labelled and highlighted in transparent colour.

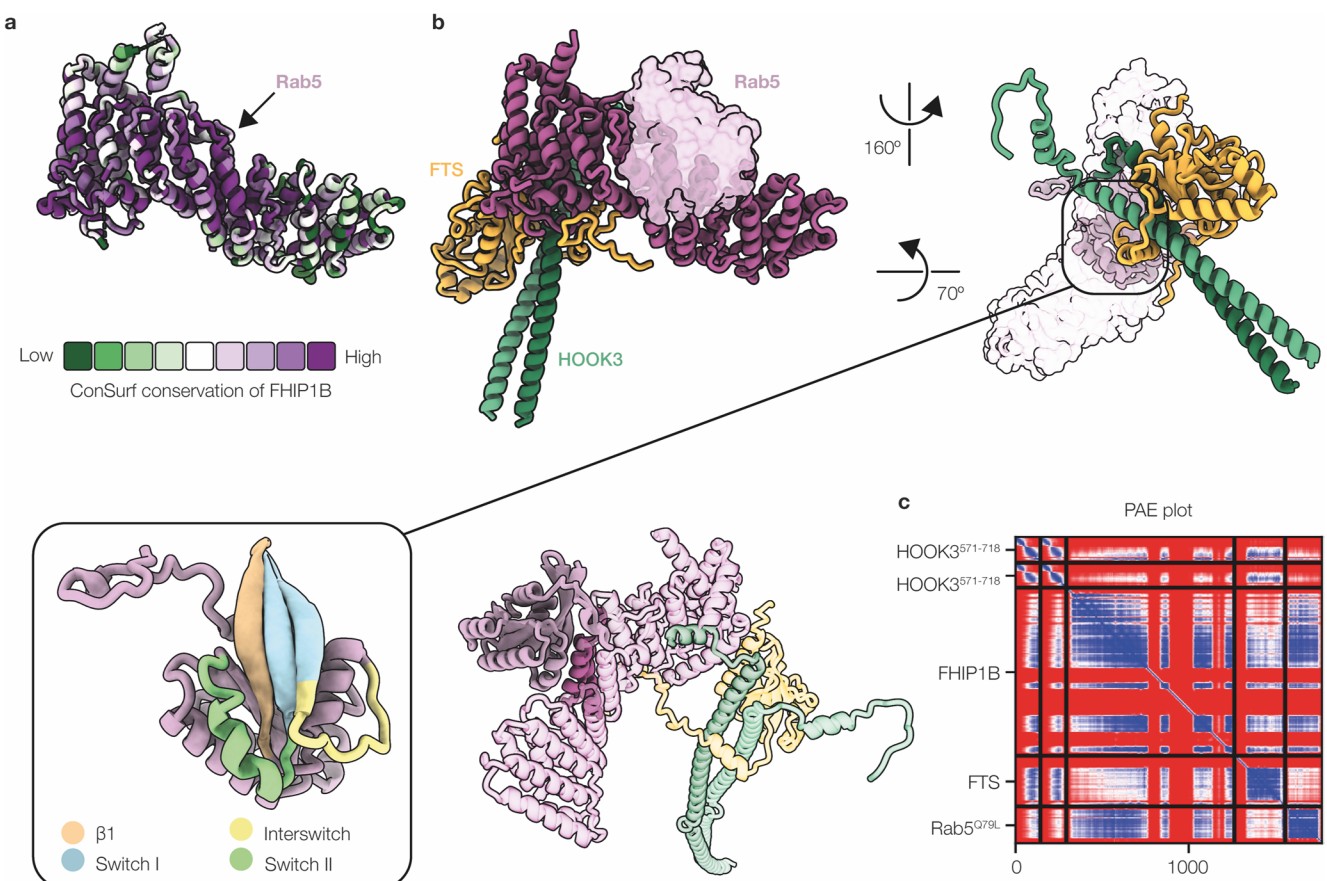

**Extended Data Fig. 4 | Conservation of FHIP1B and putative binding to Rab5. a**, ConSurf[86] evolutionary conservation profile of FHIP1B. Model of FHIP1B coloured according to level of conservation (key below) and the view of the structure is the same as that in (**b**). **b**, AlphaFold2 prediction of FH[571-718]F+Rab5[Q79L]. Predicted structure was superimposed on experimental cryo-EM FHF structure and used to show the interaction with Rab5 in three different views (top row middle and right panels, and bottom row, middle panel). Boxed inset shows predicted domain boundaries of Rab5 based on[87]. **c**, AlphaFold2 Predicted Aligned Error (PAE) plot of FH[571-718]F+Rab5[Q79L].

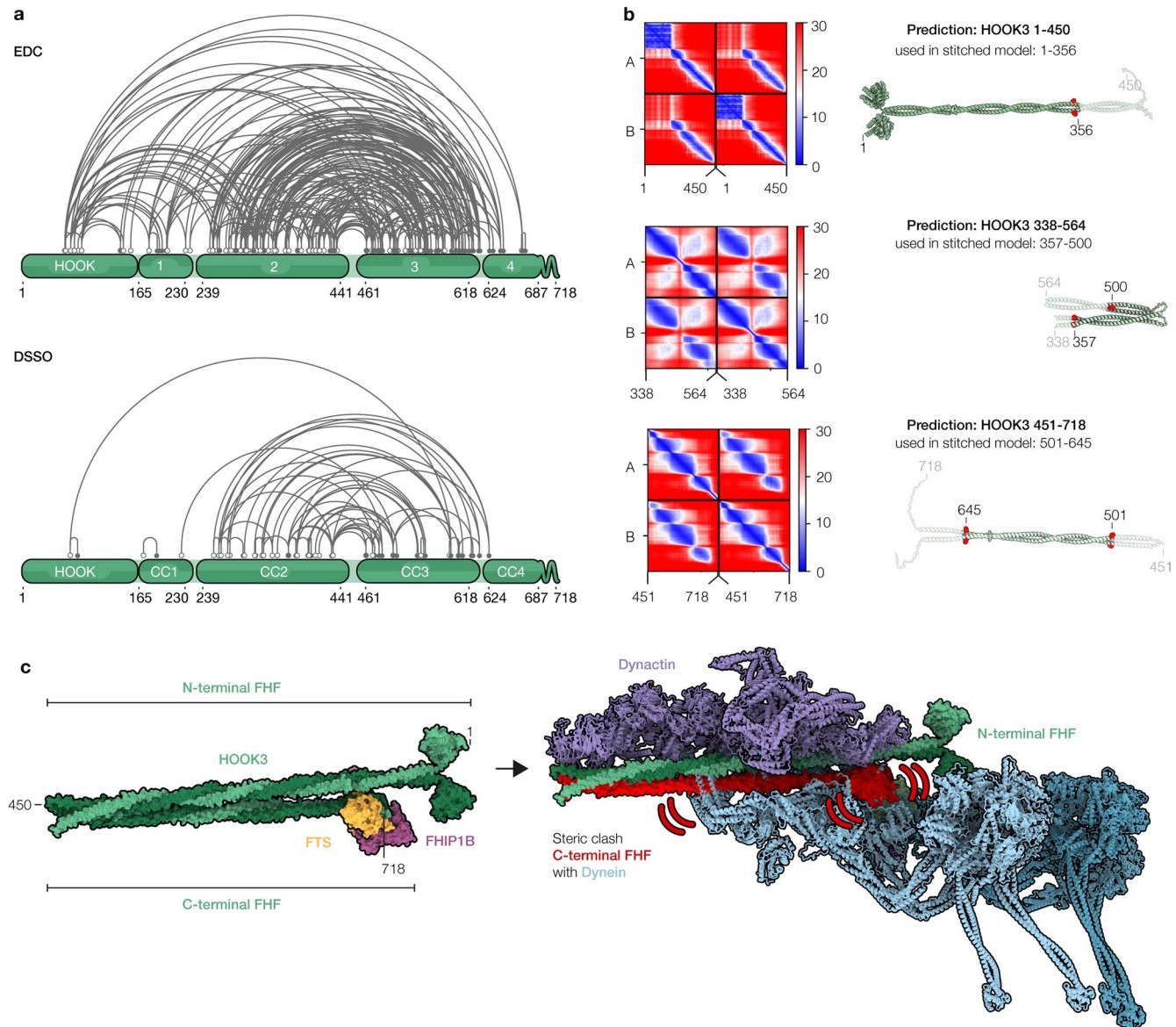

**Extended Data Fig. 5 | Crosslinking mass spectrometry and structural modelling of the auto-inhibited state of FHF. a**, Intramolecular crosslinks, using EDC (top) and DSSO (bottom), mapped on the full-length HOOK3 domain architecture. **b**, AlphaFold2 Predicted Aligned Error (PAE) plots and corresponding structural prediction of three partially overlapping HOOK3 fragments (lengths of fragments indicated in the figure). Fragments were aligned at overlapping regions using the MatchMaker tool in UCSF Chimera[88];

regions of overlap were deleted at the regions indicated by the red spheres shown on the right. **c**, The stitched full-length model of FHF (shown in surface rendering) predicts a steric clash upon binding to the dynein-dynactin complex (shown in cartoon rendering). In particular, the C-terminus of HOOK3, FTS and FHIP1B within auto-inhibited FHF (shown in red) is incompatible with stable binding to the dynein tails.

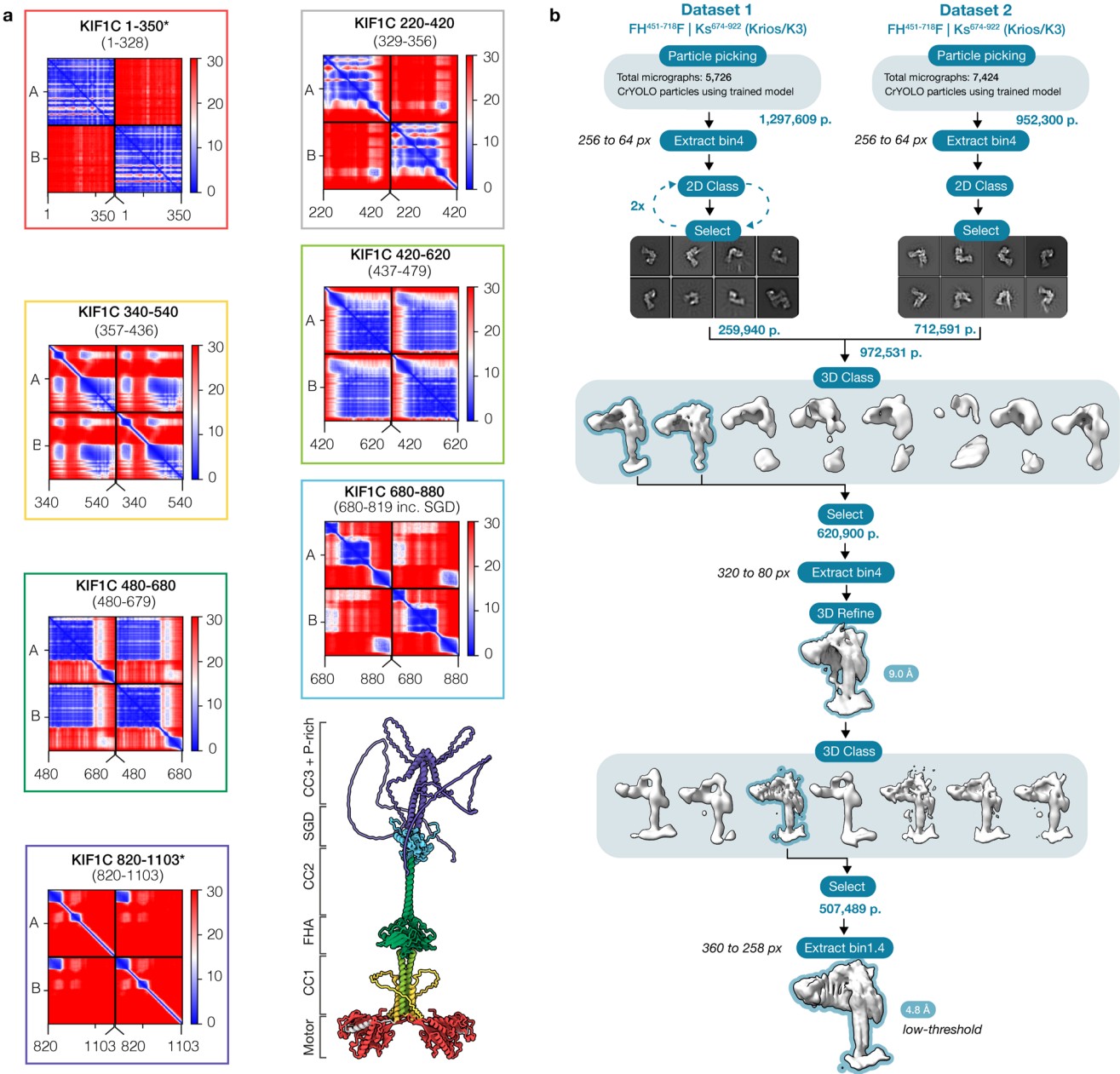

**Extended Data Fig. 6 | Structural modelling of full-length KIF1C and cryo-EM processing workflow for FH⁴⁵¹⁻⁷¹⁸F bound to KIF1C stalk (674-922).**
**a**, AlphaFold2 Predicted Aligned Error (PAE) plots of KIF1C segments used to create the stitched model of the full-length molecule. Asterisk in 1–350 prediction refers to use of the monomer motor for superposition onto the 220–420 dimer structure. Asterisk in 820–1103 prediction highlights the high level of disorder predicted for the Proline-rich region (this segment is used for completion of structure only rather than analysis). Sequences in brackets are the residues left after removal of overlapping superposed regions within the stitched molecule. **b**, Cryo-EM processing workflow for two combined FHF-KIF1C stalk datasets. RELION-4.1[69] was used throughout, apart from particle picking which was performed in crYOLO[70].

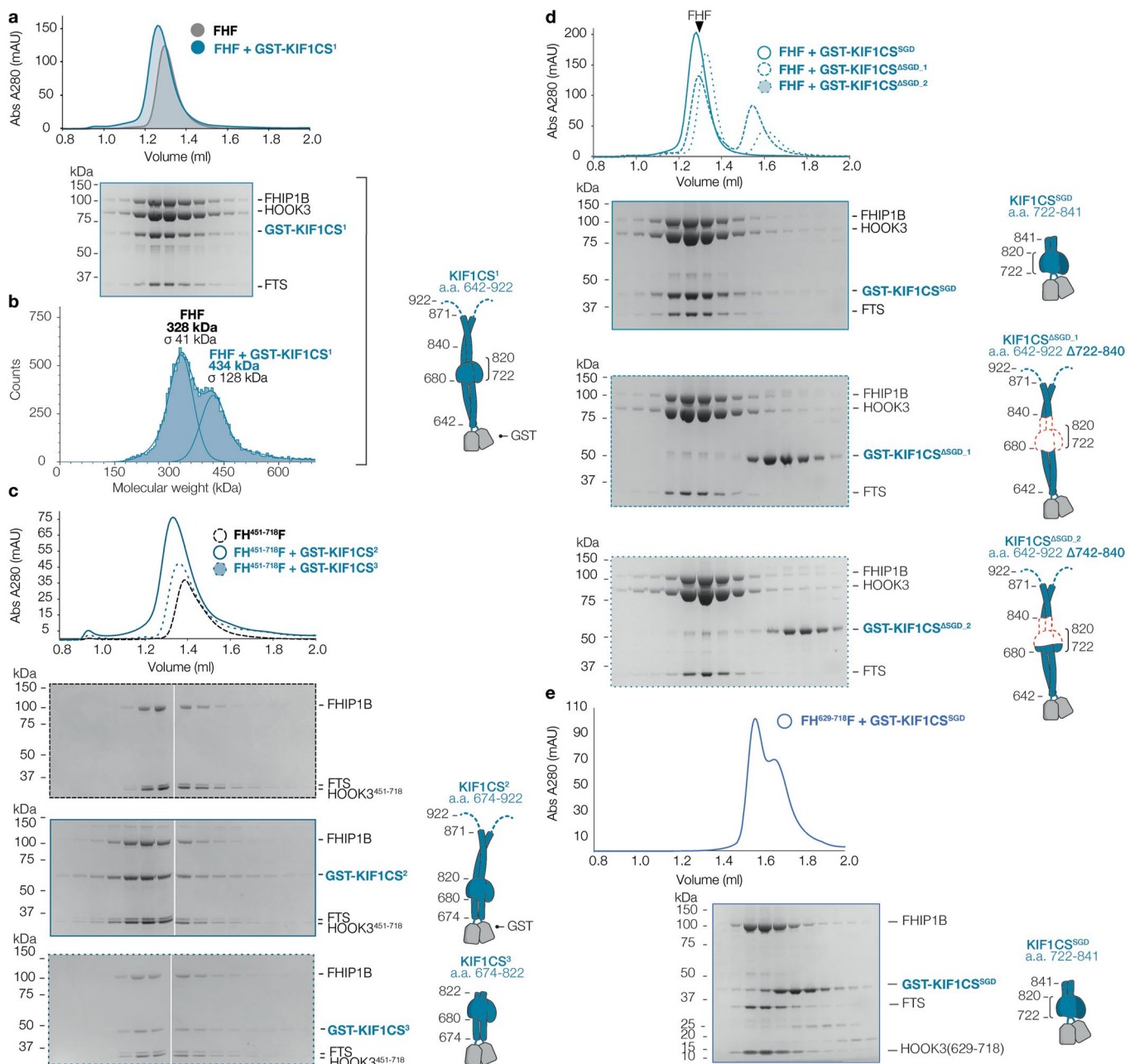

**Extended Data Fig. 7 | Biochemical characterisation of the FHF and KIF1C stalk interaction. a**, Size Exclusion Chromatography (SEC) of isolated full-length FHF and FHF + KIF1C stalk (residues 642-922). Experiment repeated three times. **b**, Mass photometry of full-length FHF and KIF1C stalk using the Refeyn machine. Cartoon on the right depicts the KIF1C stalk construct used for **a-b**. Experiment performed once. **c**, SEC reconstitution of FH[451-718]F in the absence or presence of two shorter KIF1C stalk constructs (residues 674-922 and 674-822, respectively)

depicted in the cartoons on the right. Experiment repeated twice. **d**, SEC reconstitution of full-length FHF in the presence and absence of the KIF1C stalk SGD domain. Two SGD deletion constructs within KIF1C stalk were tested (Δ722-840 and Δ742-840) against SGD construct (residues 722-841), as depicted in the cartoons on the right. Experiment performed once. **e**, SEC reconstitution of truncated FH[629-718]F with SGD construct shows no binding. Experiment performed once. SDS-PAGE gels for all runs show fractions from the peak regions.

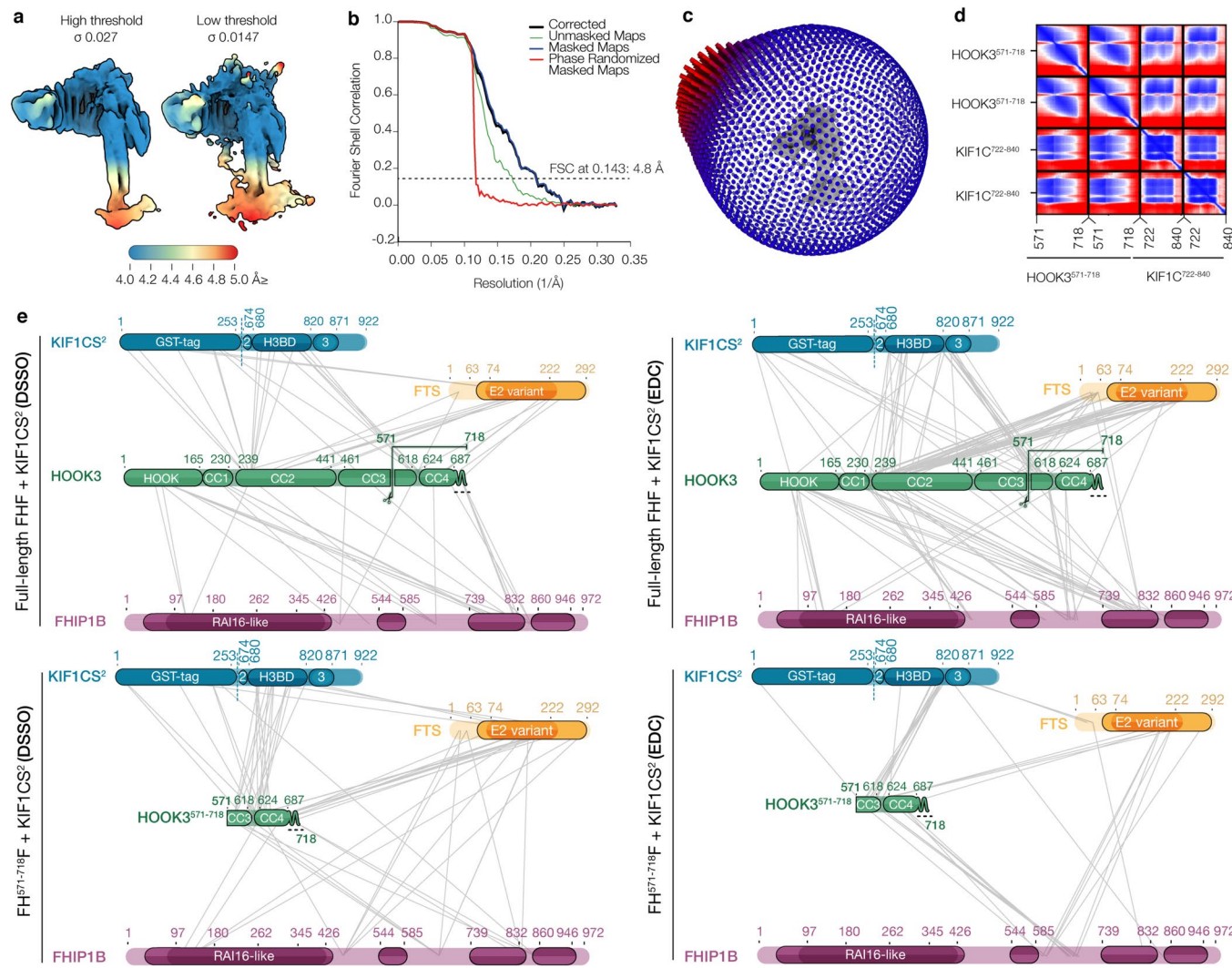

**Extended Data Fig. 8 | FHF-KIF1C cryo-EM, AlphaFold2 and cross-linking mass spectrometry validation and analysis. a**, RELION local-resolution plot applied to FH451-718F bound to KIF1CS674-922 (also referred to as KIF1CS2) cryo-EM structure, shown at two different density thresholds. **b**, Gold standard Fourier shell correlation (FSC) curves as determined by RELION-4.1 (FSC = 0.143). **c**, 3D depiction of angular distribution for the final FH451-718F-KIF1CS674-922 structure with the cryo-EM density shown in the middle. **d**, AlphaFold2 Predicted Aligned Error (PAE) plot for the HOOK3571-718-KIF1C stalk SGD722-840 prediction. **e**, Crosslinking mass spectrometry results for full-length FHF or FH571-718F bound to KIF1C stalk (containing residues 674-922, denoted KIF1CS2), either using the EDC or DSSO crosslinkers.

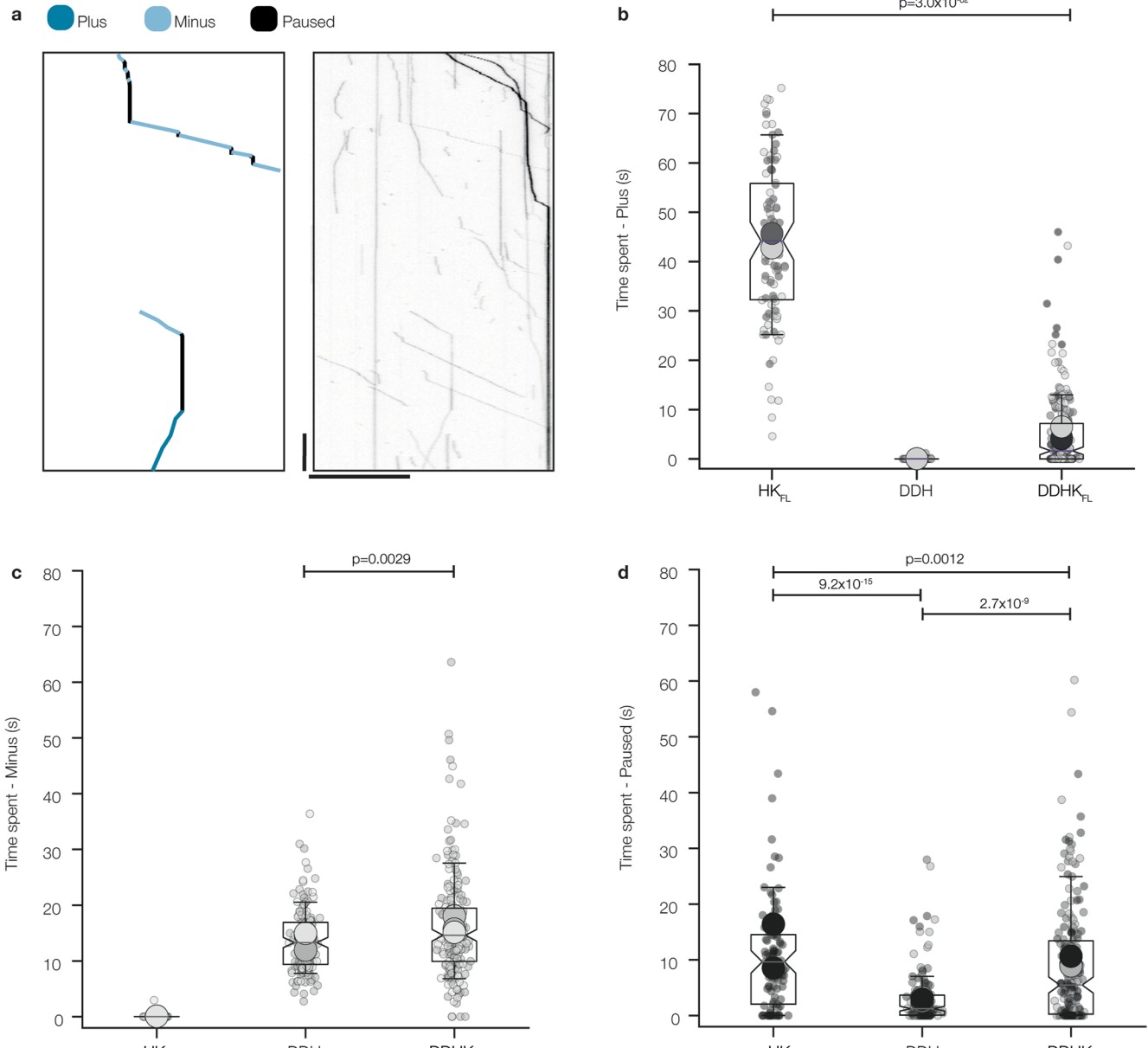

**Extended Data Fig. 9 | Subclassification of motor complex motility and pausing events. a**, Example kymograph and annotation showing what would be classified as plus and minus end directed motility and paused (moving < 25 nm/s). The example shows the TMR-dynein channel of a DDHK chamber. The scale bar is 20 μm and 20 s in the horizontal and vertical axes, respectively. **b-d**, Superplots of HK$_{FL}$, DDH and DDHK$_{FL}$ time spent moving towards the plus end (**b**), time spent moving towards the minus end (**c**) and time spent pausing at

a speed < 25 nm/s (**d**). Note that wholly static tracks (total distance <1000 nm) are excluded from this analysis. Small dots show data per microtubule and large dots show experimental averages. Boxes show quartiles with whiskers spanning 10%-90% of the data. n=94, 165 and 202 for HK$_{FL}$, DDH and DDHK$_{FL}$, respectively. Exact p values shown above graphs using Kruskal Wallis H test *used* followed by a Conover's posthoc test to evaluate pairwise interactions with a multiple comparison correction applied using Holm–Bonferroni Source data.

# Reporting Summary

## Statistics

For all statistical analyses, confirm that the following items are present in the figure legend, table legend, main text, or Methods section.

| n/a | Confirmed | |
|---|---|---|
| ☐ | ☒ | The exact sample size (*n*) for each experimental group/condition, given as a discrete number and unit of measurement |
| ☐ | ☒ | A statement on whether measurements were taken from distinct samples or whether the same sample was measured repeatedly |
| ☐ | ☒ | The statistical test(s) used AND whether they are one- or two-sided <br> *Only common tests should be described solely by name; describe more complex techniques in the Methods section.* |
| ☒ | ☐ | A description of all covariates tested |
| ☒ | ☐ | A description of any assumptions or corrections, such as tests of normality and adjustment for multiple comparisons |
| ☐ | ☒ | A full description of the statistical parameters including central tendency (e.g. means) or other basic estimates (e.g. regression coefficient) AND variation (e.g. standard deviation) or associated estimates of uncertainty (e.g. confidence intervals) |
| ☐ | ☒ | For null hypothesis testing, the test statistic (e.g. *F*, *t*, *r*) with confidence intervals, effect sizes, degrees of freedom and *P* value noted <br> *Give P values as exact values whenever suitable.* |
| ☒ | ☐ | For Bayesian analysis, information on the choice of priors and Markov chain Monte Carlo settings |
| ☒ | ☐ | For hierarchical and complex designs, identification of the appropriate level for tests and full reporting of outcomes |
| ☒ | ☐ | Estimates of effect sizes (e.g. Cohen's *d*, Pearson's *r*), indicating how they were calculated |

*Our web collection on statistics for biologists contains articles on many of the points above.*

## Software and code

Policy information about availability of computer code

| Data collection | 1. Cryo-EM data collection: EPU 2.6.1 <br> 2. Single molecule TIRF data collection: Micromanager v1.4 and xCellence |
|---|---|
| Data analysis | 1. Image processing: RELION v4.0 and wrappers within: MotionCor2, CTFFIND4 <br> 2. Particle picking: crYOLO 1.7.5 <br> 3. Model building and refinement: Coot 0.93 and PHENIX 1.14 and 1.20 <br> 4. Model and map analysis: UCSF Chimera 1.14 and UCSF Chimera X 1.4 <br> 5. Structure prediction: AlphaFold2, Colabfold 1.5.2 and AlphaScreen (https://github.com/sami-chaaban/alphascreen) <br> 6. Star file mining: StarParser 1.38 (https://github.com/sami-chaaban/starparser) <br> 7. Mass photometry: Refeyn AcquireMP and DiscoverMP, v2.3 <br> 8. Kymograph analysis: FiJi v2.9.0 <br> 9. Graphs and statistics: Prism v9.0.0 and python custom scripts <br> 10. Sequence alignments: Jalview 9.0.5 <br> 11. Conservation analysis: ConSurf web server <br> 12. Adobe Illustrator 2023 and 2024 (Adobe) |

For manuscripts utilizing custom algorithms or software that are central to the research but not yet described in published literature, software must be made available to editors and reviewers. We strongly encourage code deposition in a community repository (e.g. GitHub). See the Nature Portfolio guidelines for submitting code & software for further information.

# Data

Atomic coordinates and cryo-EM maps have been deposited in the protein data bank (PDB) and Electron Microscopy Data Bank (EMDB) under the accession codes EMD-18302 and PDB 8QAT for FHF structure and EMD-18303 for FHF + KIF1C stalk. All gel filtration, SDS-PAGE, crosslinking mass spectrometry, single molecule microscopy and AlphaFold2 raw data and associated analysis tables are deposited to Zenodo (10.5281/zenodo.10949991 and 10.5281/zenodo.11360634).

# Research involving human participants, their data, or biological material

| | |
|---|---|
| Reporting on sex and gender | n/a |
| Reporting on race, ethnicity, or other socially relevant groupings | n/a |
| Population characteristics | n/a |
| Recruitment | n/a |
| Ethics oversight | n/a |

Note that full information on the approval of the study protocol must also be provided in the manuscript.

# Field-specific reporting

Please select the one below that is the best fit for your research. If you are not sure, read the appropriate sections before making your selection.

☒ Life sciences ☐ Behavioural & social sciences ☐ Ecological, evolutionary & environmental sciences

For a reference copy of the document with all sections, see nature.com/documents/nr-reporting-summary-flat.pdf

# Life sciences study design

All studies must disclose on these points even when the disclosure is negative.

| | |
|---|---|
| Sample size | The sample size for cryo-EM was not pre-determined and depended on data collection of a first dataset (ranging between 5,000-26,000 movies depending on the time allocated on the electron microscope). If resulting structure did not yield side-chain resolution and/or an isotropic cryo-EM density then additional datasets were collected and combined. This strategy improved the resolution obtained for the FHF structure and provided additional missing views to overcome anisotropy. If adding the new data did not improve the density significantly, as observed with the FHF-KIF1C stalk datasets, then no further data were collected and cryo-EM analysis was combined with other methods to improve assignment of protein domains. The sample size for single molecule TIRF data in Figures 2 and 7 was not pre-determined. Several technical replicates were performed on three different experiment days and all usable data pooled for each day. Power analysis tests were performed for single molecule TIRF data in Figure 4 to determine how many technical replicates to perform. 15 microtubules per technical n for FHF-containing samples was determined to be feasible based on the average concentration of microtubules in the field of view. |
| Data exclusions | Exclusions for cryo-EM analysis included ice "particles", carbon edges, junk/non-averaged protein density and low-resolution/broken density classes during particle sorting stages (2D and 3D classification). For analysis of TIRF data in Figure 4, any events that were stationary or moved <1.2s duration or <500 nm were discounted and not classed in the processive event or run length classification. For analysis of TIRF data in Figures 1, 2 and 7, data were only analysed from single microtubules that showed exclusively unidirectional runs after KIF1C-GFP flow-in. Runs were defined as detailed in the methods. |
| Replication | Cryo-EM data findings were replicated through the collection of 5 separate and distinct datasets (from different FHF preparations), revealing an identical architecture and organization of FHF. FHF-containing TIRF datasets were repeated at least four times (often using proteins from distinct purifications [biological replicates]), each time showing the same trend of results. |
| Randomization | For calculation of the gold-standard FSC, cryo-EM particles were randomly split into two halves using RELION. |
| Blinding | Blinding is not relevant for structural studies here as grouping is not applicable to image processing. For FHF-containing single molecule studies, blinding would not be appropriate as the conditions showed a clear phenotype that would be straightforward to distinguish even in the absence of sample labelling. For TIRF assays shown in Figures 1, 2 and 7, all single microtubules were included in kymograph analysis and |

all motors were traced in kymographs. Filtering of runs and measurements was extracted via scripts in an unbiased manner, making blinding obsolete.

# Reporting for specific materials, systems and methods

We require information from authors about some types of materials, experimental systems and methods used in many studies. Here, indicate whether each material, system or method listed is relevant to your study. If you are not sure if a list item applies to your research, read the appropriate section before selecting a response.

| Materials & experimental systems | | Methods | |
|---|---|---|---|
| n/a | Involved in the study | n/a | Involved in the study |
| ☒ | ☐ Antibodies | ☒ | ☐ ChIP-seq |
| ☐ | ☒ Eukaryotic cell lines | ☒ | ☐ Flow cytometry |
| ☒ | ☐ Palaeontology and archaeology | ☒ | ☐ MRI-based neuroimaging |
| ☒ | ☐ Animals and other organisms | | |
| ☒ | ☐ Clinical data | | |
| ☒ | ☐ Dual use research of concern | | |
| ☒ | ☐ Plants | | |

## Eukaryotic cell lines

Policy information about cell lines and Sex and Gender in Research

| | |
|---|---|
| Cell line source(s) | Sf9 cells were used for expression and purification of HOOK3, FHF, dynein, KIF1C-GFP and Lis1 (ThermoFisher Scientific cat. no. 11496015). Dynactin was purified from pig brains. KIF1C stalk constructs were expressed and purified from E.coli SoluBL21 cells (Bio Cat GMB Cat. No. C700200-GL). |
| Authentication | No cell line authentication was used. |
| Mycoplasma contamination | Cells were not tested for mycoplasma contamination. |
| Commonly misidentified lines (See ICLAC register) | No commonly misidentified cell lines were used. |

## Plants

| | |
|---|---|
| Seed stocks | n/a |
| Novel plant genotypes | n/a |
| Authentication | n/a |

