## [Peer Review File · Nature Structural & Molecular Biology]

KIF1C activates and extends dynein movement through the FHF cargo adaptor

Corresponding Author: Professor Anne Straube

Version 0:

Decision Letter:

28th Nov 2023

Dear Professor Straube,

Thank you again for submitting your manuscript "KIF1C activates and extends dynein movement through the FHF cargo adaptor". I apologise for the delay in responding, which resulted from the difficulty in obtaining suitable referee reports. Nevertheless, we now have comments (below) from the 3 reviewers who evaluated your paper. In light of these reports, we remain interested in your study and would like to see your response to the comments of the referees, in the form of a revised manuscript.

You will see that though the reviewers appreciate the novelty of the findings, they raise several concerns that need to be addressed in a revised manuscript. More specifically, all experts point to the need for further discussion, better data presentation, figure additions/reorganisation, resolving discrepancies to help the reader. More importantly, all experts find that alternative or parallel scenarios could be plausible and thus ask for toning down certain conclusions or adequately discussing such alternatives. Along these lines, reviewer #2 requests expanding the pull-down assays so as to lend further credit to proposed mechanisms and reviewer #3 poses a very important technical question (points 13-14) that must be addressed.

Please be sure to address/respond to all concerns of the referees in full in a point-by-point response and highlight all changes in the revised manuscript text file. If you have comments that are intended for editors only, please include those in a separate cover letter.

We expect to see your revised manuscript within 3 months. If you cannot send it within this time, please contact us to discuss an extension; we would still consider your revision, provided that no similar work has been accepted for publication at NSMB or published elsewhere.

Reporting Summary:

If there are additional or modified structures presented in the final revision, please submit the corresponding PDB validation reports and the maps/coordinates for such structures.

Data availability: this journal strongly supports public availability of data. All data used in accepted papers should be available via a public data repository, or alternatively, as Supplementary Information. If data can only be shared on request, please explain why in your Data Availability Statement, and also in the correspondence with your editor. Please note that for some data types, deposition in a public repository is mandatory - more information on our data deposition policies and available repositories can be found below:

<https://www.nature.com/nature-research/editorial-policies/reporting-standards#availability-of-data>

Link Redacted

Sincerely,

Dimitris Typas
Associate Editor
Nature Structural & Molecular Biology
ORCID: 0000-0002-8737-1319

Reviewers' Comments:

Reviewer #1:

Remarks to the Author:

The manuscript by Ali and coworkers reports that the plus-end-directed microtubule (MT)-associated motor protein KIF1C not only activates the minus-end-directed MT motor complex, dynein-dynactin-HOOK (DDH), but also surprisingly increases its processivity (run length). Using cryo-EM, cross-linking experiments, and AlphaFold modeling, the authors further demonstrate that KIF1C activates DDH motion by relieving the auto-inhibitory conformation of the cargo-adaptor HOOK in complex with its adaptor complex FHS. The reported results will be of significant interest to the cell biological and

cytoskeletal motor communities. Since the study is carried out with great care and rigor, I have only a few minor comments:

1. When coupling the opposing motors KIF1C and DDH together, the authors observe only a 2% rate of direction reversals. It would be valuable for the authors to discuss the reasons behind the rarity of direction reversals.

2. On page 2, in the second paragraph, the authors write, "This suggests that coupling opposite polarity motors via their adaptors supports processive unidirectional transport and prevents a tug-of-war." Considering that direction reversals seem to occur after an elongated pause (see Fig. 1C), I believe the data actually suggest that direction reversals follow a tug-of-war. If KIF1C and DDH are both trying to move in opposite directions with similar force-generating capabilities, one can imagine that a few steps in the plus and minus-end direction occur during a tug-of-war process before either KIF1C or DDH wins. As the resolution limit of the used microscope is only ~250 nm, even tens of ~8 nm center-of-mass steps could occur in both directions without the authors noticing it, which could explain the observed pauses between minus-end- and plus-end-directed motion. The authors should discuss this possibility. In the absence of a tug-of-war, the direction reversals should be instantaneous (considering the temporal aspects of protein diffusion and rearrangements of motor domains).

3. In the text on page 2, the authors write that they observed 16% of plus-end-directed motion and refer to Fig. 1b. However, Fig. 1b shows a bar of only 13%.

Reviewer #2:

Remarks to the Author:

In the manuscript by Abid Ali et al., the authors set out to understand the interplay between dynein and kinesin motility, as governed by the shared activating adaptor, Hook3. In brief, the authors demonstrate the following: (1) Addition of Kif1C promotes microtubule binding/landing of dynein-dynactin (DD), even without the Kif1C motor domain; (2) Addition of DD promotes microtubule binding/landing of Kif1C, even without the dynein motor domain (in fact, dynactin is sufficient); (3) A cryoEM structure of FHIP1B + FTS + Hook3C-term shows how Hook3 is critical for FHF complex formation, and also reveals that the Rab5 binding site is still exposed in assembled FHF complex (i.e., the site is fully accessible); (4) Hook3 bound to FHIP1B-Fts is still an autoinhibited adaptor, but addition of Kif1C stalk activates DD landing, indicating that Kif1C can activate motility in the presence of a full FHF complex; (5) X-link/MS data combined with AlphaFold2 confirms that Hook3 indeed adopts an autoinhibited conformation; (6) Using a combination of cryoEM, AF2 predictions, Xlink/MS, and pull-downs, the authors map the interactions between Kif1C (to its stalk globular domain, a.k.a., the "HBD") and FHF (to a region of Hook3 that is non-overlapping with FHIP1B + Fts, suggesting they do not impact each other's binding); (6) Using pull-down assays, they confirm that the Hook3 C-term and N-term indeed interact, and that Kif1C prevents this interaction; (7) They find that Kif1C and dynein are likely engaged – albeit weakly – with the microtubule while the opposite polarity motor is engaged in processive movement.

Based on their findings, the authors propose a combination of mechanisms by which kinesin and dynein rely on each other for transport in cells: a steric disinhibition model (whereby each motor somehow relieves the autoinhibited state of their counterpart), and a microtubule tethering model (whereby a weakly attached motor acts as a processivity factor for the motor that is engaged in processive transport). They suggest that their steric disinhibition model may indeed apply to other adaptors since e.g. BicDR1 and BicD2 similarly engage with dynein and kinesin via their N- and C-termini, respectively. In brief, the paper is very well written, the figures are easy to follow, and the data are of very high quality. I highly recommend publication after the authors address a few minor concerns, as detailed below.

1) The authors focus on the mechanism by which Kif1C binding to Hook3 relieves its autoinhibition, thus promoting its interaction with DD. However, given the findings presented in Figure 1 (that DtD promotes Kif1C landing), wouldn't the converse also be true? Specifically, is it possible that DD binding to Hook3 also relieves Hook3 autoinhibition, and thus promotes Kif1C-Hook3 binding? If so, their pull-down assay presented in Figure 3F could be used to test this. For example, the authors could include DtD with the FHcF-Hn mixture, and this might potentially disrupt Hc-Hn binding. I realize this might complicate the SDS-PAGE analysis given the inclusion of the additional proteins, so perhaps the authors could use an alternate means to test this idea (e.g., immunoblot)? This is not a requisite for publication, but would address an important question raised by the data shown in Figure 1.

2) Similarly, the model presented in Figure 5 is ignoring the fact that DD can also promote Kif1C transport. I wonder if their model should include an alternate pathway for cargo transport activation in which dynein-dynactin are recruited to FHF prior to Kif1C. This would theoretically also promote kinesin-transport.

3) I think Extended Data Fig. 10, along with the associated paragraph (the one centered around line 250) can be moved to the Results section.

4) With regard to Extended Data Fig. 2: It is unclear why tubulin is present on the top-left SDS-PAGE for panel A. I didn't notice any microtubule pelleting step for the purification in the methods section. It's a minor point, but I found this curious.

Reviewer #3:

Remarks to the Author:

The authors combine single-molecule tracking, cryo-EM, and biochemical approaches to investigate how the activating

adapter protein Hook3 activates both kinesin and dynein motors to form a bidirectional complex. This work builds nicely on previous work on KIF1C by the Straube lab, and foundational cryoEM work of activated dynein-dynactin complexes by the Carter lab. The finding that both motors are activated by a shared adapter has important implications for understanding bidirectional cargo transport in neurons; namely, that we can think about pairs of motors that can be (co)regulated as opposed to imagining that there are varying numbers of kinesins and dyneins on the cargo that are operating independently.

The data are strong, there are interesting new structures, and most importantly the structural data and the single-molecule results dovetail very nicely to tell the story. The writing is generally clear, but it is dense, and the figures are generally good, but a bit packed and lacking details in some places (described below). Overall, this is a strong contribution to the field that should shift the thinking about how kinesin and dynein motors are regulated during bidirectional transport.

Comments:

1. Line 47: "complexes undertook processive minus end (dynein-driven, 80%) or plus end-directed transport (kinesin-driven, 16%) (Fig. 1a-b)". In Figure 1b, the graph shows 83% and 13% respectively. This should be fixed to avoid confusion.
2. Figure 1: It would really help the reader to show a diagram of the motility in panel A, like the diagrams in panel D. Particularly since it is first panel in paper, a bit of a roadmap would help.
3. Do the authors have a speculation as to why full length Kif1C doesn't recruit the same amount of dynein as the kif1Cstalk in Fig 1d? Do the heads have some sort of inhibitory effect?
4. Why are they called processive events in Fig. 2E but landing rate in Fig 1E and 1G? These are the same measurements presumably.
5. Line 64: 'dynein microtubule recruitment' is sort of an awkward phrase.
6. Figure 2: In this paper it's not always obvious which structures are cryoEM and which are alpha fold, etc. Maybe a small 'AF' symbol next to the structures would help.
7. Figure 2d: The authors should consider moving the motor complex up away from the microtubule in the middle panel (DDFHF) to denote lack of binding.
8. Line 116: "We found that processive dynein-dynactin-FHF (DDFHF) complexes were formed 11-fold less efficiently than DDH complexes made with a constitutively active truncated HOOK3 construct (DDH1-522)". In Fig 1, landing rate was used to measure activation level, whereas here landing rate is used to assess assembly efficiency. It seems formally possible that the lower landing rate here is from an inhibited complex and not from lack of assembly. Can the authors discern between these two possibilities?
9. Figure 2D: How do we know that this DDHFL complex is inhibited in the complex or if the FHF is preventing the complex from forming and that's why there are few dynein landing events? Dynein/Dynactin without an adaptor are mostly in solution so it's possible if the FHF is autoinhibited in solution there might just be reduced complexes formed. If the complex is formed on MT and autoinhibited on the MT then we should potentially see static dyneins not just a reduction in dynein events. And in Fig 2D panel 2, what does that mean for the motile events, is the inhibition sometimes relieved? I think that promotes the idea that the full length FHF prevents complex formation but for the ones that actually form the complex they move fine.
10. It would be helpful to add data DDHK points from figure 1 to Figure 2E to help the comparison of different complexes.
11. Figure 3F: Perhaps we missed this, but if lane 1 is a load and not a pull down you should make it clear since it seems only HookN is present in that one.
12. Figure 4: This is a pretty brutal figure that is hard to take in. Here's a suggestion. First, put the run time data in supplementary – that is least important and it is just RL/vel anyway. Second, combine the plus and minus RL plots into one with a positive and negative y-axis, and do the same with velocity. That will reduce it to two plots with basically the same information.
13. Line 461: 'Microtubules prepared in this way were kept for 3-4 weeks with the tube wrapped in aluminium foil, and diluted 1:50 in taxol-containing buffer prior to use in microscopy chambers'
14. Could MT concatenation cause any issues in their experiments? 3-4 weeks is a long time to store MT, which can concatenate in a few days.

Minor issues:

Figure 1: A lot of good information but cartoons are a little small to see and gets overwhelming to understand all of them at once.

Figure 2: the cartoons above the kymographs are way too small to see.

Line 437: 'Coverslips were stored between layers of lens tissue (Ross Optical, AG806) inside pipette boxes until they were used'. Was there a maximum storage time?

Figure 3A: it would be better to color code the regions and residues according to the cartoon in 3C

Version 1:

Decision Letter:

Our ref: NSMB-A48394A

21st Jun 2024

Dear Professor Straube,

Thank you for submitting your revised manuscript "KIF1C activates and extends dynein movement through the FHF cargo adaptor" (NSMB-A48394A). It has now been seen by the original referees and their comments are below. The reviewers find that the paper has improved in revision, and therefore we are happy to accept it in principle in Nature Structural & Molecular Biology, pending minor revisions to satisfy the referees' final requests and to comply with our editorial and formatting guidelines.

We are now performing detailed checks on your paper and will send you a checklist detailing our editorial and formatting requirements in about 2-3 weeks. I apologise for this slightly extended timeline but our editorial team is disrupted by editorial absences at the moment. Please do not upload the final materials and make any revisions until you receive this additional information from us.

To facilitate our work at this stage, it is important that we have a copy of the main text as a word file. If you could please send along a word version of this file as soon as possible, we would greatly appreciate it; please make sure to copy the NSMB account (cc'ed above).

Thank you again for your interest in Nature Structural & Molecular Biology. Please do not hesitate to contact me if you have any questions.

Sincerely,

Dimitris Typas
Senior Editor
Nature Structural & Molecular Biology
ORCID: 0000-0002-8737-1319

Reviewer #1 (Remarks to the Author):

The authors have satisfactorily addressed my comments. The revisions have improved the manuscript, and all major issues have been resolved. I recommend accepting the manuscript for publication.

Reviewer #2 (Remarks to the Author):

The authors have done an excellent job addressing my concerns, as well as those of the other reviewers. I am in full support of publication of the revised manuscript.

Reviewer #3 (Remarks to the Author):

The authors have fully responded to all of our comments, and made good arguments for suggestions they chose not to take. Nice work.

Version 2:

Decision Letter:

3rd Oct 2024

Dear Professor Straube,

We are now happy to accept your revised paper "KIF1C activates and extends dynein movement through the FHF cargo adaptor" for publication as an Article in Nature Structural & Molecular Biology.

Acceptance is conditional on the manuscript's not being published elsewhere and on there being no announcement of this work to the newspapers, magazines, radio or television until the publication date in *Nature Structural & Molecular Biology*.

Over the next few weeks, your paper will be copyedited to ensure that it conforms to *Nature Structural & Molecular Biology* style. Once your paper is typeset, you will receive an email with a link to choose the appropriate publishing options for your paper and our Author Services team will be in touch regarding any additional information that may be required.

Your paper will be published online soon after we receive proof corrections and will appear in print in the next available issue. You can find out your date of online publication by contacting the production team shortly after sending your proof corrections.

Please note that *Nature Structural & Molecular Biology* is a Transformative Journal (TJ). Authors may publish their research with us through the traditional subscription access route or make their paper immediately open access through payment of an article-processing charge (APC). Authors will not be required to make a final decision about access to their article until it has been accepted. <https://www.springernature.com/gp/open-research/transformative-journals> Find out more about Transformative Journals

Authors may need to take specific actions to achieve <https://www.springernature.com/gp/open-research/funding/policy-compliance-faqs> compliance with funder and institutional open access mandates. If your research is supported by a funder that requires immediate open access (e.g. according to <https://www.springernature.com/gp/open-research/plan-s-compliance> Plan S principles) then you should select the gold OA route, and we will direct you to the compliant route where possible. For authors selecting the subscription publication route, the journal's standard licensing terms will need to be accepted, including <https://www.springernature.com/gp/open-research/policies/journal-policies> self-archiving policies. Those licensing terms will supersede any other terms that the author or any third party may assert apply to any version of the manuscript.

Sincerely,

Dimitris Typas
Senior Editor
Nature Structural & Molecular Biology
ORCID: 0000-0002-8737-1319

KIF1C activates and extends dynein movement through the FHF cargo adaptor.

Responses to Reviewers' Comments

Reviewer #1:

Remarks to the Author:

The manuscript by Ali and coworkers reports that the plus-end-directed microtubule (MT)-associated motor protein KIF1C not only activates the minus-end-directed MT motor complex, dynein-dynactin-HOOK (DDH), but also surprisingly increases its processivity (run length). Using cryo-EM, cross-linking experiments, and AlphaFold modeling, the authors further demonstrate that KIF1C activates DDH motion by relieving the auto-inhibitory conformation of the cargo-adaptor HOOK in complex with its adaptor complex FHS. The reported results will be of significant interest to the cell biological and cytoskeletal motor communities. Since the study is carried out with great care and rigor, I have only a few minor comments:

1. When coupling the opposing motors KIF1C and DDH together, the authors observe only a 2% rate of direction reversals. It would be valuable for the authors to discuss the reasons behind the rarity of direction reversals.

We thank the reviewer for this great suggestion. We have now added our thoughts on the higher occurrence of directional switches of cargoes in cells versus in these reconstitution experiments to the discussion. One possibility is that the apparently higher directional switch frequency in cells is due to the more complex organisation of the underlying microtubule network where antiparallel microtubules might be more prevalent and therefore even a single motor that switches tracks could drive directional reversal of cargoes. In our experiments here, we carefully control for observing only motility on uniform filaments by flow-in of KIF1C-GFP at the end of each experiment to check microtubule polarity and remove any tracks that show bidirectional movement of kinesin alone. Motor complexes might also be more prone to directional switches in cells due to obstructions acting on the cargo, which in turn might stall the driving motor and trigger the directional change. Such obstacles could be the end of the microtubule, tight crossovers of another microtubule or other cytoskeletal filament, defects in the microtubule lattice, MAPs and other motors on the microtubule. In addition to such mechanical effects, biochemical cues such as binding of regulatory factors and posttranslational modifications could trigger switch frequencies in a spatially controlled manner. Likewise, the turnover of motors on the cargo could cause directional switches. The reconstituted system we established, will allow probing those mechanisms in the future and thereby our understanding how the presence of two opposite polarity motors is advantageous for efficient long-distance cargo transport in cells.

The discussion of this can now be found on page 10-11 lines 324-329.

2. On page 2, in the second paragraph, the authors write, "This suggests that coupling opposite polarity motors via their adaptors supports processive unidirectional transport and prevents a tug-of-war." Considering that direction reversals seem to occur after an elongated pause (see Fig. 1C), I believe the data actually suggest that direction reversals follow a tug-of-war. If KIF1C and DDH are both trying to move in opposite directions with similar force-generation capabilities, one can imagine that a few steps in the plus and minus-end direction occur during a tug-of-war process before either KIF1C or DDH wins. As the resolution limit of the used microscope is only ~250 nm, even tens of ~8 nm center-of-mass steps could occur

in both directions without the authors noticing it, which could explain the observed pauses between minus-end- and plus-end directed motion. The authors should discuss this possibility. In the absence of a tug-of-war, the direction reversals should be instantaneous (considering the temporal aspects of protein diffusion and rearrangements of motor domains).

KIF1C-driven motility events frequently contain pauses even in the absence of dynein, while dynein tends to pause rarely in the absence of KIF1C (see Extended Data Fig. 9d). DDHK complexes show intermediate pause times. We don't know what causes KIF1C pause states and whether this is a strongly bound state that could also trigger pauses in DDHK complexes or a weakly bound state that might explain the dominance of dynein-driven motion in the DDHK complexes. However, we cannot fully exclude that some of the pauses are phases of force balance between KIF1C and dynein. However, if that was the primary cause of pauses, we would probably expect a higher occurrence of directional switches as symmetry breaking after a period of force balance should not consistently favour the direction before pause. We have now added some of these thoughts to the discussion on page 11 lines 336-339.

3. In the text on page 2, the authors write that they observed 16% of plus-end directed motion and refer to Fig. 1b. However, Fig. 1b shows a bar of only 13%.

Thank you for spotting this error, we have now corrected this.

Reviewer #2:

Remarks to the Author:

In the manuscript by Abid Ali et al., the authors set out to understand the interplay between dynein and kinesin motility, as governed by the shared activating adaptor, Hook3. In brief, the authors demonstrate the following: (1) Addition of Kif1C promotes microtubule binding/landing of dynein-dynactin (DD), even without the Kif1C motor domain; (2) Addition of DD promotes microtubule binding/landing of Kif1C, even without the dynein motor domain (in fact, dynactin is sufficient); (3) A cryoEM structure of FHIP1B + FTS + Hook3C-term shows how Hook3 is critical for FHF complex formation, and also reveals that the Rab5 binding site is still exposed in assembled FHF complex (i.e., the site is full accessible); (4) Hook3 bound to FHIP1B-Fts is still an autoinhibited adaptor, but addition of Kif1C stalk activates DD landing, indicating that Kif1C can activate motility in the presence of a full FHF complex; (5) X-link/MS data combined with AlphaFold2 confirms that Hook3 indeed adopts an autoinhibited conformation; (6) Using a combination of cryoEM, AF2 predictions, Xlink/MS, and pull-downs, the authors map the interactions between Kif1C (to its stalk globular domain, a.k.a., the "HBD") and FHF (to a region of Hook3 that is non-overlapping with FHIP1B + Fts, suggesting they do not impact each other's binding); (6) Using pull-down assays, they confirm that the Hook3 C-term and N-term indeed interact, and that Kif1C prevents this interaction; (7) They find that Kif1C and dynein are likely engaged – albeit weakly – with the microtubule while the opposite polarity motor is engaged in processive movement.

Based on their findings, the authors propose a combination of mechanisms by which kinesin and dynein rely on each other for transport in cells: a steric disinhibition model (whereby each motor somehow relieves the autoinhibited state of their counterpart), and a microtubule tethering model (whereby a weakly attached motor acts as a processivity factor for the motor that is engaged in processive transport). They suggest that their steric disinhibition model may indeed apply to other adaptors since e.g. BicDR1 and BicD2 similarly engage with dynein and kinesin via their N- and C-termini, respectively. In brief, the paper is very well

written, the figures are easy to follow, and the data are of very high quality. I highly recommend publication after the authors address a few minor concerns, as detailed below.

1) The authors focus on the mechanism by which Kif1C binding to Hook3 relieves its autoinhibition, thus promoting its interaction with DD. However, given the findings presented in Figure 1 (that DtD promotes Kif1C landing), wouldn't the converse also be true? Specifically, is it possible that DD binding to Hook3 also relieves Hook3 autoinhibition, and thus promotes Kif1C-Hook3 binding? If so, their pull-down assay presented in Figure 3F could be used to test this. For example, the authors could include DtD with the FHcF-Hn mixture, and this might potentially disrupt Hc-Hn binding. I realize this might complicate the SDS-PAGE analysis given the inclusion of the additional proteins, so perhaps the authors could use an alternate means to test this idea (e.g., immunoblot)? This is not a requisite for publication, but would address an important question raised by the data shown in Figure 1.

We thank the reviewer for this great suggestion. We performed experiments test dynein-mediated activation of FHF, but the assay was not reliable due to the inherently low dynein tail and dynactin concentrations we can obtain. For FH^CF and H^N to interact stably, high micromolar concentrations of these proteins are required. We confirmed this by performing fluorescence polarisation assays as shown below (experiment performed by Stephen McLaughlin from the MRC-LMB Biophysics facility). This showed FH^CF and H^N to have a K_d of 32 μ M fitting within the bounds of 27 to 39 μ M at a confidence limit of 95 % using physiological salt concentrations (150 mM NaCl).

Figure 1: Fluorescent anisotropy measurement with 25 nM NT495 labelled-HOOK3(1-450) and titrating a complex of HOOK3(451-718)-FHIP1B-FTS. The binding curve gave a calculated K_d of 32 μ M [27 μ M to 39 μ M at 95% CI].

We used 5 μ M concentration of Strep-FH^CF and 10 μ M untagged H^N using low salt conditions (50 mM KCl) in the original pull-down assay and observed binding between these two components (new Figure 6b-c). When performed at lower concentrations that allowed adding equimolar concentrations of dynein tail and dynactin, we could not observe binding of FH^CF and H^N and could therefore not test whether dynein could activate FHF.

2) Similarly, the model presented in Figure 5 is ignoring the fact that DD can also promote Kif1C transport. I wonder if their model should include an alternate pathway for cargo transport activation in which dynein-dynactin are recruited to FHF prior to Kif1C. This would theoretically also promote kinesin-transport.

This has now been implemented: we depict autoinhibited KIF1C and DD with arrows pointing to their HOOK3 binding sites in the first step of the model (Fig. 8a), the original KIF1C-mediated activation of dynein (Fig. 8b) and the vice versa scenario (Fig. 8c).

3) I think Extended Data Fig. 10, along with the associated paragraph (the one centered around line 250) can be moved to the Results section.

We have implemented this suggestion and moved the data showing that our FHF structure applies to other combinations of HOOK and FHIP proteins also into the main text on page 4 and Figure 3d-g and describe this together with the HOOK-FTS-FHIP1B cryo-EM structure.

4) With regard to Extended Data Fig. 2: It is unclear why tubulin is present on the top-left SDS-PAGE for panel A. I didn't notice any microtubule pelleting step for the purification in the methods section. It's a minor point, but I found this curious.

We established by mass spectrometry that insect cell tubulin co-purifies with recombinant full-length human FHF in our preparations (but not in instances where the HOOK3 N-terminus is truncated [compare the right and left panels of Extended Data Fig 2a]). This observation is apparent without introducing additional microtubule pelleting steps. This finding is consistent with Walenta et al. (2001), where they show that the HOOK3 N-terminus (but not the C-terminus) directly interacts with microtubules (Figure 7B in their paper). SEC-MALS (Extended Data Fig 2a) and mass photometry (Extended Data Fig 7b) suggest that in solution, this is a sub-stoichiometric tubulin interaction as the major FHF molecular species is only compatible with a dimer of HOOK3 bound to FHIP1B and FTS. We do not consider HOOK3 microtubule binding as significant for this paper to warrant inclusion in the discussion because the relevant region would not be able to reach down and interact with the microtubule when bound to dynein and KIF1C.

Reviewer #3:

Remarks to the Author:

The authors combine single-molecule tracking, cryo-EM, and biochemical approaches to investigate how the activating adapter protein Hook3 activates both kinesin and dynein motors to form a bidirectional complex. This work builds nicely on previous work on KIF1C by the Straube lab, and foundational cryoEM work of activated dynein-dynactin complexes by the Carter lab. The finding that both motors are activated by a shared adapter has important implications for understanding bidirectional cargo transport in neurons; namely, that we can think about pairs of motors that can be (co)regulated as opposed to imagining that there are varying numbers of kinesins and dyneins on the cargo that are operating independently.

The data are strong, there are interesting new structures, and most importantly the structural data and the single-molecule results dovetail very nicely to tell the story. The writing is generally clear, but it is dense, and the figures are generally good, but a bit packed and lacking details in some places (described below). Overall, this is a strong contribution to the field that should shift the thinking about how kinesin and dynein motors are regulated during bidirectional transport.

Comments:

1. Line 47: "complexes undertook processive minus end (dynein-driven, 80%) or plus end-

directed transport (kinesin-driven, 16%) (Fig. 1a-b)". In Figure 1b, the graph shows 83% and 13% respectively. This should be fixed to avoid confusion.

We thank the reviewer for spotting this error. This has now been fixed.

2. Figure 1: It would really help the reader to show a diagram of the motility in panel A, like the diagrams in panel D. Particularly since it is first panel in paper, a bit of a roadmap would help.

We implemented this suggestion and added diagrams to Figure 1.

3. Do the authors have a speculation as to why full length Kif1C doesn't recruit the same amount of dynein as the kif1Cstalk in Fig 1d? Do the heads have some sort of inhibitory effect?

We have shown intramolecular autoinhibitory interactions between the KIF1C motor domain and the HOOK3 binding region in KIF1C previously. While HOOK3 can activate KIF1C, it must compete with the motor domain and we found that HOOK3 only resulted in 2-fold activation of full length KIF1C, while KIF1C without the inhibitory stalk region was about 10-fold more active (Siddiqui, Zwetsloot et al. 2019). Therefore, it is reasonable to expect that the stalk region alone is also about 10-fold more efficient in activating FHF as there is no competition with the motor domains. We have added this notion to the text on page 10 lines 286-290.

4. Why are they called processive events in Fig. 2E but landing rate in Fig 1E and 1G? These are the same measurements presumably.

These are not the same measurements because landing rate data in Fig. 2 show all events that have been traced even if these were completely stationary. For data in previous Figure 2 and new Figure 4, static and diffusive events are not included and processive events were defined as 1.2 s duration on MTs and moved >500 nm.

5. Line 64: 'dynein microtubule recruitment' is sort of an awkward phrase.

Agreed, this has now been rephrased to "recruitment of DDH to microtubules" on page 2 line 57.

6. Figure 2: In this paper it's not always obvious which structures are cryoEM and which are alpha fold, etc. Maybe a small 'AF' symbol next to the structures would help.

We have now added labels in the figure panels of Fig. 4-6 to indicate which structures show cryo-EM-based models and which are based on AlphaFold and/or cross-linking mass spectrometry data.

7. Figure 2d: The authors should consider moving the motor complex up away from the microtubule in the middle panel (DDFHF) to denote lack of binding.

FHF is a much less efficient dynein activator if compared to truncated HOOK3 N-terminal region, but DDFHF is a highly processive motor complex when it forms. To show this more clearly, we have now included DD as a control and include both the frequency of processive events and the run length distributions in Figure 4b-c.

8. Line 116: "We found that processive dynein-dynactin-FHF (DDFHF) complexes were

formed 11-fold less efficiently than DDH complexes made with a constitutively active truncated HOOK3 construct (DDH1-522)". In Fig 1, landing rate was used to measure activation level, whereas here landing rate is used to assess assembly efficiency. It seems formally possible that the lower landing rate here is from an inhibited complex and not from lack of assembly. Can the authors discern between these two possibilities?

We have modified the text to make clear that we do not directly measure complex assembly, but initiation of processive transport events here.

9. Figure 2D: How do we know that this DDHFL complex is inhibited in the complex or if the FHF is preventing the complex from forming and that's why there are few dynein landing events? Dynein/Dynactin without an adaptor are mostly in solution so it's possible if the FHF is autoinhibited in solution there might just be reduced complexes formed. If the complex is formed on MT and autoinhibited on the MT then we should potentially see static dyneins not just a reduction in dynein events. And in Fig 2D panel 2, what does that mean for the motile events, is the inhibition sometimes relieved? I think that promotes the idea that the full length FHF prevents complex formation but for the ones that actually form the complex they move fine.

We agree and have now included a dynein-dynactin control kymograph and data to illustrate that dynein/dynactin is landing on microtubules but is only weakly processive in line with previously published data (Schlager et al. EMBO J 2014; Trokter et al. PNAS 2012). FHF activates dynein but is less efficient than a truncated and constitutively active HOOK3. Because DDFHF complexes are highly processive, we consider it unlikely that the presence of HOOK3 C-terminus and FHIP/FTS interactors inhibit or revert dynein to its folded once complexes are formed. We have now expanded the text to explain this in more detail on page 5.

10. It would be helpful to add data DDHK points from figure 1 to Figure 2E to help the comparison of different complexes.

Including a dataset with full-length KIF1C and HOOK3 in the absence of FHIP and FTS to the data in Fig. 4a-b would have required repeating all the datasets under identical conditions for comparison (Fig. 1-2, 6 vs Fig. 4 single molecule studies were performed in different institutes with different microscopes). We did not consider this essential. However, we felt it was important to add a DD control (for which we had data obtained in parallel). To aid comparison, we are also now making statements highlighting the relative activation observed between DD and DDH, and DD and DDFHF as well as DDH and DDHKs, and DDFHF and DDFHFKs in the main text on pages 2, 3 and 5.

11. Figure 3F: Perhaps we missed this, but if lane 1 is a load and not a pull down you should make it clear since it seems only HookN is present in that one.

This is the negative control to detect unspecific binding of H^N to the beads. We have now included the input gels and showed an overview of which constructs are included (Fig. 6). We hope this facilitates a more intuitive understanding of the experiment shown.

12. Figure 4: This is a pretty brutal figure that is hard to take in. Here's a suggestion. First, put the run time data in supplementary – that is least important and it is just RL/vel anyway. Second, combine the plus and minus RL plots into one with a positive and negative y-axis, and do the same with velocity. That will reduce it to two plots with basically the same information.

When we look at minus end directed motility, we compare three different complexes that contain full length dynein. When we look at plus end directed motility, we look at four different complexes containing full length KIF1C. Therefore, we look at different complexes with only the full DDHK appearing in both sets and cannot combine the plus and minus end-directed data as suggested. We also consider it useful to show the dwell time data to illustrate that primarily changes in dwell time drive the extended runs of dynein in the presence of full-length KIF1C, while dynein has no impact on the dwell time of KIF1C. This would be more difficult to grasp with undertaking the suggested run length / speed. To aid understanding of the complexes analysed in both parts of the figure, we do now show cartoons of plus and minus end-directed complexes separately.

13. Line 461: 'Microtubules prepared in this way were kept for 3-4 weeks with the tube wrapped in aluminium foil, and diluted 1:50 in taxol-containing buffer prior to use in microscopy chambers'

14. Could MT concatenation cause any issues in their experiments? 3-4 weeks is a long time to store MT, which can concatenate in a few days.

Microtubule concatenation is commonly used in the field to obtain very long microtubule substrates for transport assays, which are difficult to obtain through freshly assembled preparations. Because tubulin can exchange in the lattice and defects in the microtubule shaft can be repaired, concatenated microtubules should be considered a suitable substrate for these motility assays with immobilised microtubules. We are confident of this because we checked every microtubule from which we analysed motility of motor complexes that it is unipolar and continuous by flowing in KIF1C-GFP at the end of each experiment.

Minor issues:

Figure 1: A lot of good information but cartoons are a little small to see and gets overwhelming to understand all of them at once.

We have decompressed the figures and split previous Figure 1 into two figures, which allowed adding additional schematics and enlarging some of the panels.

Figure 2: the cartoons above the kymographs are way too small to see.

This is now enlarged in Figure 4a.

Line 437: 'Coverslips were stored between layers of lens tissue (Ross Optical, AG806) inside pipette boxes until they were used'. Was there a maximum storage time?

Usage rates were high when we collected the final datasets included in the manuscript, so coverslips were usually used within a week and maximally stored for one month. We also wish to note that plasma cleaning was always performed on the day of the experiment just before assembling the imaging chambers.

Figure 3A: it would be better to color code the regions and residues according to the cartoon in 3C

Previous Figure 3A and 3B show KIF1C which is in blue. The structural model in previous 3C (now Fig. 5C) shows the C-terminal portion of HOOK3, full-length FHIP1B and FTS in the presence of KIF1C stalk. All are coloured according to the consistent colour scheme in the paper (for example domain architecture now in new Figure 3a). Because of this we didn't deem

it necessary to repeat the domain architectures again especially as the major focus in old figure 3C and new figure 5C is on the KIF1C stalk extra density. However, we added labels to each of the proteins shown.